# Climate Change and Mycotoxins Trends in Serbia and Croatia: A 15-Year Review

**DOI:** 10.3390/foods13091391

**Published:** 2024-04-30

**Authors:** Jovana Kos, Bojana Radić, Tina Lešić, Mislav Anić, Pavle Jovanov, Bojana Šarić, Jelka Pleadin

**Affiliations:** 1Institute of Food Technology in Novi Sad, University of Novi Sad, Bulevar Cara Lazara 1, 21000 Novi Sad, Serbia; bojana.radic@fins.uns.ac.rs (B.R.); pavle.jovanov@fins.uns.ac.rs (P.J.); bojana.saric@fins.uns.ac.rs (B.Š.); 2Laboratory for Analytical Chemistry, Croatian Veterinary Institute, Savska Cesta 143, 10000 Zagreb, Croatia; lesic@veinst.hr (T.L.); pleadin@veinst.hr (J.P.); 3Croatian Meteorological and Hydrological Service, Ravnice 48, 10000 Zagreb, Croatia; mislav.anic@cirus.dhz.hr

**Keywords:** mycotoxins, food, Serbia, Croatia, climate change

## Abstract

This review examines the 15-year presence of mycotoxins in food from Serbia and Croatia to provide a comprehensive overview of trends. Encompassing the timeframe from 2009 to 2023, this study integrates data from both countries and investigates climate change patterns. The results from Serbia focus primarily on maize and milk and show a strong dependence of contamination on weather conditions. However, there is limited data on mycotoxins in cereals other than maize, as well as in other food categories. Conversely, Croatia has a broader spectrum of studies, with significant attention given to milk and maize, along with more research on other cereals, meat, and meat products compared to Serbia. Over the investigated 15-year period, both Serbia and Croatia have experienced notable shifts in climate, including fluctuations in temperature, precipitation, and humidity levels. These changes have significantly influenced agriculture, consequently affecting the occurrence of mycotoxins in various food products. The results summarized in this 15-year review indicate the urgent need for further research and action to address mycotoxins contamination in Serbian and Croatian food supply chains. This urgency is further emphasized by the changing climatic conditions and their potential to exacerbate public health and food safety risks associated with mycotoxins.

## 1. Introduction

Mycotoxins, secondary metabolites produced by toxigenic mold, represent a global unpredictable and unavoidable threat. The presence of mycotoxins affects the quality and safety of food and feed, and thus the economy through losses in food and feed production and healthcare costs. Exposure to mycotoxins can have consequences for human and animal health, such as mutagenicity, teratogenicity, carcinogenicity, and immunosuppression. These effects can further lead to various health problems, including liver and kidney damage, immune system disorders, and even death in severe cases [1,2,3]. Several hundred different compounds have been isolated worldwide and chemically classified as mycotoxins, while only about 50 of them have been studied in detail [2]. The classification of mycotoxins into regulated, unregulated, and emerging is based precisely on the knowledge of their impact on health. Regulated mycotoxins, including aflatoxins (AFB1, AFB2, AFG1, AFG2, and AFM1), deoxynivalenol (DON), zearalenone (ZEN), ochratoxin A (OTA), patulin (PAT), and fumonisins (FB1 and FB2), are those for which regulations, restrictions, and maximum levels (MLs) in certain food and feed exist. On the other hand, unregulated mycotoxins, including tenuazonic acid (TeA), alternariol (AOH), alternariolmethylether (AME), tentoxin, and sterigmatocystin (STC), and emerging mycotoxins, including moniliformin (MON), enniatins (ENNs), beauvericin (BEA), fusaproliferin (FP), and many others, lack specific regulatory limits. Their presence and impact are still being studied [4,5]. Various factors are known to influence the production of mycotoxins, including environmental factors, biological factors, agricultural practices, food and feed distribution, and processing conditions [2,3,6]. However, the latest research, as an ongoing global concern, increasingly emphasizes the impact of climate change on the production of various mycotoxins [6,7,8,9,10].

The widespread occurrence of mycotoxins has been documented throughout many scientific articles, while one of the most cited references where the Food and Agriculture Organization (FAO) [11], refers to the fact that 25% of the world’s cereals are contaminated with mycotoxins. However, Eskola et al. [12] conducted a comprehensive study on the validity of this widely cited “FAO estimate” of 25%. After analysis, the authors concluded that the origin of this statement is unknown as there does not appear to be an accurate published reference for the figure of 25% of the world’s crops infected with mycotoxins, nor is there any information on what data set this estimate is based on or how it was calculated. After discussions between Eskola et al. [12] and FAO officials, it appeared that neither had been able to trace the origin of the estimate. By reviewing the literature, Eskola et al. [12] concluded that the current mycotoxins contamination above the European Union (EU) and Codex limits confirms the FAO estimate of 25%, while this percentage is much higher (60–80%) for the total contaminated cereals. Such a high and increased percentage of cereal contamination can be explained, on one hand, by the improved sensitivity of the analytical methods used for mycotoxins analysis, and on the other hand, by the impact of more frequent weather extremes and climate change [6,13,14].

In both Serbia and Croatia, improvements in the use of analytical methods for mycotoxins analysis and climate change have been observed in recent years. Over the past 15 years, extensive research has been conducted in Croatia and Serbia regarding the presence of mycotoxins in maize and milk [9,15,16,17,18,19,20,21,22,23,24], while much less attention has been paid to other cereals and cereal products, in which only a few studies are available on the occurrence of mycotoxins in other foods, primarily meat products in Croatia [25,26,27,28,29,30,31,32,33,34,35].

In the initial part of the mentioned 15-year period, the majority of published articles from Serbia and Croatia utilized the ELISA (Enzyme-linked immunosorbent assay) technique for the analysis of mycotoxins presence in different food samples. Later, other techniques were also incorporated and more frequently applied. Briefly, there has been a gradual transition to high-performance liquid chromatography (HPLC), a confirmatory technique using a photodiode array (DAD) or fluorescence (FLD) detectors. Furthermore, in the last decade, the HPLC technique has often been combined with tandem mass spectrometry (MS/MS) for mycotoxins analysis. In both countries, Croatia and Serbia, due to the lack of LC-MS/MS, investigations [22,23,24,36,37,38,39,40] have been realized abroad, mainly in Austria, using methods developed by Malachová et al. [41] and Sulyok et al. [13]. Subsequently, LC-MS/MS methods have been developed and utilized in both Serbia and Croatia [9,10,42,43], but they mostly focus on a smaller number of mycotoxins compared to those used in Austria. Based on the available studies, it can be noticed that there are not many multi-mycotoxins methods developed in Serbia or Croatia or in neighboring countries.

All of the aforementioned points highlight the crucial importance of controlling mycotoxins for both public health and economic advancement within the country. Therefore, numerous strategies for the control of mycotoxins have been considered in different areas of the world, including European countries. The EU boasts the world’s most comprehensive and meticulous regulations concerning mycotoxins in food [44] and feed [45,46], indicative of a profound commitment to their prevention and control. In Croatia, the Regulations about MLs in food and feed are fully aligned with the EU. On the other hand, it is important to highlight that certain discrepancies characterize the regulations on mycotoxins in food [47] and feed in Serbia [48] compared to those in the EU. Briefly, regarding foodstuffs, there are certain differences between EU and Serbian Regulations for AFM1 and citrinin (CIT). The ML for AFM1 in milk is established at 0.25 µg/kg by the Serbian Regulation [47], whereas the EU mandates a more stringent ML of 0.050 µg/kg [44]. CIT is not prescribed in the Serbian Regulation [47]. Regarding, the presence of mycotoxins in food for infants and small children, in Serbia, it is prescribed by the Regulations on the health suitability of dietary products [49] which is harmonized with the EU directives for that area. Regarding feedstuffs, currently, MLs are defined in the EU only for AFB1, while guidance levels are set for DON, ZEN, OTA, and fumonisins (FUMs). Serbian regulations [48] set an ML of 30 µg/kg for AFB1 in feed materials, whereas the EU regulation establishes a more stringent limit of 20 µg/kg [46]. The allowed levels for ZEN in feed materials, as well as ZEN and OTA in complementary and complete feed stuffs, are twice as high in Serbian Regulation. In addition, in Serbia, there is no guidance level for the presence of FUMs in animal feed, although there are data on their presence both in maize and animal feed mixtures. Furthermore, in the EU, indicative levels have been introduced for T-2 and HT-2 toxins in cereals and cereal products [50], which are not prescribed in Serbia. Compliance with the aforementioned regulations is important for preventing the spread of mycotoxins and protecting human and animal health. It is also important to emphasize that the available regulations in Europe and the world do not take into account the co-occurrence of multiple mycotoxins, while cereals are very often contaminated with several different mycotoxins.

The aim of this review paper is to present an overview of the results published from Serbia and Croatia regarding the occurrence of mycotoxins in different food matrices (cereals, milk and dairy products, meat and meat products, and other food) over the past 15 years, from 2009 to 2023. In addition, given that weather conditions significantly contribute to the occurrence of mycotoxins in food, a detailed analysis of weather indicators for the period 2009–2023 in comparison to long-term period (1981–2010) was conducted, highlighting the trend of climate change in these two countries. Serbia and Croatia were chosen together because they are two neighboring countries with similar weather conditions, agricultural practices, and dietary habits of the population. To the best of our knowledge, this is the first review paper that combines fifteen years of results from two different countries.

## 2. Climate Trends in Serbia and Croatia

Serbia is situated on the Balkan Peninsula and generally has a moderate continental climate with distinct local characteristics. In the northern and central regions, the climate is more continental, characterized by cold winters and hot, humid summers with well-distributed rainfall. In the south, summers and autumns are typically drier, while winters are relatively cold, often accompanied by heavy inland snowfall in the mountain regions. According to the Köppen climate classification, the northern part of Serbia is characterized by a humid subtropical (Cfa) and hot-summer humid continental climate (Dfa), while the climate of the southern part of Serbia mainly belongs to a warm-summer humid continental climate (Dfb) [51,52].

The climate of Croatia is determined by its location in the northern temperate latitudes and the associated large- and medium-scale weather processes. The most important modifiers of the climate in Croatia are the Adriatic Sea and the Mediterranean Sea, the orography of the Dinarides, the openness of the north-eastern parts to the Pannonian Plain, and the diversity of vegetation [53]. According to the Köppen classification, the coastal part of Croatia is primarily characterized by a hot-summer Mediterranean climate (Csa). The mountain zone has a warm-summer humid continental climate (Dfb), whereas the lowland areas of Croatia, in addition to the warm-summer humid continental climate (Dfb), also experience a predominantly humid subtropical climate (Cfa) [51,54].

Observed climate changes in Southeastern Europe, encompassing Serbia and Croatia, over the past decades, have resulted in significant temperature variations and alterations in precipitation patterns. These changes are marked by rising air temperatures and more frequent occurrences of drought conditions during the summer months, along with sporadic instances of extremely high precipitation. Such trends towards warmer and drier conditions have already started to affect public health, human well-being, agricultural productivity, forest ecosystems, and various other aspects of the environment and society. The most notable climate changes in Serbia as well as in Croatia have been observed over the past two decades [52,54,55]. The average air temperature trend across the territory of Serbia during the period from 1961 to 2017 was approximately 0.36 °C per decade. However, from 1981 to 2017, this trend of temperature increase was 0.60 °C per decade. Values of the SPI (standardized precipitation index) index for August for the territory of Serbia during the period 1950–2017 indicate a higher frequency of drought occurrence after the year 2000.

To ilustrate the trends of climate change in Serbia and Croatia from 2009 to 2023, the following figures are presented. Figure 1 and Figure 2 present meteorological conditions in Serbia (North and Central), showcasing the average air temperature and total precipitation compared to the long-term averages (1981–2010) during the growing seasons of various agricultural plants (April–September) (Figure 1), as well as monthly air temperature deviations (a) and the standardized precipitation index (b) from May to September (Figure 2). Similarly, Figure 3 and Figure 4 show analogous meteorological conditions but for lowland Croatia. The areas selected for analysis are North and Central Serbia as well as lowland Croatia because these regions have the most extensive agricultural production, and the highest number of published studies on mycotoxins in food originate from these regions.

As shown in Figure 1, throughout all seasons during the 2009–2023 period in Serbia, the mean air temperature exceeded the long-term average, with the 2012 season standing out notably. In 2012, the seasonal air temperature deviated by 2.3 °C from the long-term average, marking a particularly significant departure. The monthly air temperature anomalies (Figure 2a) for June, July, August, and September of the 2012 season were notably high, with deviations of 2.5 °C, 3.5 °C, 2.8 °C, and 3.7 °C, respectively. It is particularly noteworthy that according to Figure 2a, deviations in air temperature during the studied period of May–June 2009–2023 exceeded 2.5 °C in three months in 2012, 2018, 2022, and 2023; in two months during 2011, 2017, 2019, and 2020; and in one month in 2013 and 2021. Conversely, during the 2014 season, the smallest deviations from the average air temperatures were recorded. In terms of precipitation, the driest season occurred in 2012, while the rainiest was in 2014. According to the standardized precipitation index (SPI-2), exceptional drought was observed in July 2015, while in September 2011, September 2012, and July 2022, drought reached extreme levels (Figure 2b). In contrast, May of both 2014 and 2019 experienced extremely wet conditions.

In Croatia, the average air temperature tends to be slightly lower, while precipitation levels are higher in the lowlands compared to Northern and Central Serbia. During the 2010 and 2014 seasons, the air temperature in lowland Croatia remained consistent with the average. However, in the remaining seasons, air temperature exceeded the long-term average (1981–2010), as illustrated in Figure 3. The 2018 season was notable as the warmest season, with the air temperature surpassing the long-term average by 2.0 °C. Monthly air temperature anomalies are depicted in Figure 4a. Exceptionally rainy seasons occurred in 2010 and 2014, with the total precipitation significantly exceeding the multi-year average. Additionally, the 2019 season also experienced a notably high amount of precipitation. The 2011 season was identified as the driest, aligning with the conditions observed in Serbia. According to the SPI-2 index, extremely rainy conditions were noted in June 2010, May and September 2014, and May 2019. In contrast, extremely dry conditions were observed in September 2019 and August 2012, while severe drought was recorded in September 2009, July 2015, and July 2021.

According to different climate scenarios, an increase in air temperature is expected in both Serbia and Croatia in future periods. According to the Representative Concentration Pathway 4.5 (RCP4.5) scenario for Serbia, the mean annual temperature is projected to increase by approximately 1.5 °C during the period 2046–2065 compared to the reference period of 1986–2005. In the case of the Representative Concentration Pathway 8.5 (RCP8.5) scenario, the increase in air temperature is more pronounced. Seasonal analyses indicate that changes in the mean air temperature suggest a potentially slightly lesser increase during the colder part of the year compared to the warmer part of the year during the first half of the 21st century. In the future period, no significant trend in mean annual precipitation is anticipated for the territory of Serbia. The decrease in total precipitation during the June–August period is expected to persist in future periods according to both scenarios, RCP4.5 and RCP8.5. By the end of the 21st century, under the RCP8.5 scenario, the average precipitation decrease in the territory of Serbia is projected to be 20.5%, with a much more significant decrease in the southern regions, reaching up to 40% [52,56].

A consistent warming trend has also been observed in Croatia since the second half of the 20th century, ranging from 0.2 °C per decade along the Adriatic coast to 0.5 °C per decade in lowland areas. This increase in air temperature has led to a greater accumulation of heat and an extension of the growing season across the country. Climate projections indicate that the upward trend in air temperature is anticipated to continue in the future. According to the RCP 4.5 scenario, during the period 2041–2070, an average increase in air temperature between 1.5 °C and 1.7 °C is projected in Croatia. The findings regarding the precipitation trend highlight a clear seasonality in the observed changes. A weak precipitation trend has been noted on an annual basis, while the most significant negative trend in precipitation, ranging from −5% to −15% per decade compared to the 1981–2010 average, was observed during the summer months along the Adriatic coast and its hinterland. The total annual precipitation (RCP4.5 scenario) for the period 2041–2070 exhibits relatively minor, spatially variable changes compared to the reference period 1981–2010. The changes in the total amount of precipitation during the seasons in the period 2041–2070 vary across Croatia, with a notable decrease in precipitation during the summer throughout the entire area. In contrast, there is a prevailing, albeit less pronounced, increase in precipitation during other seasons [54].

## 3. Mycotoxin Occurrence in Serbia and Croatia

### 3.1. Maize

Maize stands out as a primary cereal crop, being the most extensively cultivated cereal both in Serbia and Croatia. In Serbia, maize cultivation is mainly concentrated in the northern and central regions, while in Croatia, it is primarily grown in the continental region. The majority, comprising approximately 40% of the total planted area of field crops in Serbia, is devoted to maize cultivation. The average maize production in Serbia, over the last 15 years, is 6.4 Mt (ranging from 3.7 to 8.1 Mt). In recent years, Serbia has been marked as one of the prominent maize exporters in Europe as well as globally, with an estimated annual export volume of around two million tons. In Serbia, only 8–10% of arable land is irrigated. Given the increasingly frequent warming trend and insufficient precipitation, particularly during the summer months, maize yields are heavily reliant on weather conditions throughout the growing season [57,58].

Croatia also has excellent soil and climatic conditions for maize. Croatia has been ranked 45th within the group of 153 countries followed in terms of maize production. Maize production was 2.24 Mt in Croatia in 2021, according to FAOSTAT [11]. Furthermore, during the year 2023, 268.1 thousand ha of maize was produced [59].

Of all agricultural crops grown in Serbia and Croatia, climate change has the greatest impact on maize. On the one hand, it affects maize yields, while on the other hand, it contributes to an increased occurrence of mycotoxins. In both countries, the maize cultivation season typically lasts from early April to the end of September. Published data indicate that the weather conditions during the summer months have the greatest impact on both yield and mycotoxin contamination, particularly since maize is predominantly grown under non-irrigated conditions in both countries [6,9,10,22,60].

#### 3.1.1. Mycotoxins in Maize from Serbia

In Serbia, between 80% and 90% of the maize produced for domestic use is allocated for feed production, with a smaller fraction utilized for food, seeds, or other industries. It is noteworthy that maize consumption for feed has been declining for decades due to a decrease in livestock numbers [58]. The findings regarding the occurrence of mycotoxins in Serbian maize, spanning from 2009 to the present, indicate an increasing trend of maize contamination with mycotoxins. This phenomenon may be correlated with the differences in analytical methods used for maize analysis, as well as the weather conditions recorded during maize cultivation. Namely, the results published between 2010 and 2017 were predominantly obtained using the ELISA method [16,61,62], while in that period, the use of the HPLC-DAD/FLD method [63] was less prevalent. The results obtained using the LC-MS/MS method from the mentioned period are not available. As a consequence, the majority of the results published during that period were focused on the presence of one or just a few different mycotoxins.

In the first study examining the presence of aflatoxins (AFS) in maize from Serbia over a multi-year period, Kos et al. [62] investigated the occurrence of aflatoxins in maize harvested between 2009 and 2012. The results obtained using the ELISA method indicated that none of the maize samples in 2009, 2010, or 2011 were contaminated with aflatoxins. However, in contrast, as many as 69% of samples in 2012 contained aflatoxins in concentrations exceeding the limit of quantification (LOQ) of the applied method, set at 1 µg/kg. The ranges and the mean determined concentrations of aflatoxins were 1.01–86.1 µg/kg and 36.3 µg/kg, respectively. The high level of maize contamination with aflatoxins during the year 2012, as well as the absence of aflatoxins in the years 2009–2011, was attributed to weather conditions by the authors. Weather conditions during the 2009 and 2010 years were characterized as moderate, without extreme events. In 2011, certain months experienced drought conditions, while in 2012, drought persisted continuously for several months, leading to an extended period of dry weather and significantly higher deviations in air temperatures. The elevated level of maize contamination with aflatoxins as well as other *Aspergillus* metabolites in Serbia during 2012 was further confirmed in studies conducted by Kos et al. [22] and Janić Hajnal et al. [36], utilizing a multi-method developed by Malachova et al. [41]. In those studies, the authors examined the presence of regulated and unregulated mycotoxins, their derivatives, as well as other fungal metabolites in maize samples from Serbia collected during the years marked by extreme drought (2012), hot and dry conditions (2013 and 2015), and extreme precipitation (2014). The findings of this study revealed that aflatoxins were detected in 94% of maize samples in 2012, with concentrations ranging from 0.6 to 205 µg/kg. Additionally, some *Aspergillus* metabolites (including kojic acid (98%), 3-nitropropionic acid (98%), averufin (96%), versicolorin C (96%), versicolorin A (84%), and O-methylsterigmatocystin (61%)) were present at exceptionally high levels in maize in 2012. In 2013 and 2015, aflatoxins were detected in 33% and 90% of samples, respectively, but at significantly lower concentrations compared to 2012, ranging from 0.5 to 48 µg/kg and 0.4 to 41 µg/kg, respectively. Conversely, aflatoxins were not detected in 2014. The maize growing season in Serbia during 2014 was characterized by an exceptionally high amount of precipitation, with certain months experiencing the maximum recorded values since meteorological observations began in Serbia. The extreme precipitation led to 100% contamination of maize with DON, ZEN, and their derivatives ZEN-sulfate and DON-3-glucoside. Additionally, alpha-ZEN, beta-ZEN, and 15-acetyl-DON were also frequently detected, with contamination frequencies ranging from 61% to 98%. Despite the varying weather conditions during maize growing seasons in the 2012–2015 period—ranging from extremely dry to extremely wet—FB1 was consistently present in 100% of the samples from each year. FB2 occurred with a frequency of 98–100%, followed by other fumonisins: FB3 in 92–100% of samples, FB4 in 96–100%, FA1 in 82–98%, and FA2 in 76–98%. Janić Hajnal et al. [36] reported that maize samples in 2012, 2013, 2014, and 2015 contained a range of 23–66, 20–47, 18–57, and 17–57 different fungal metabolites, including regulated mycotoxins, their derivatives, non-regulated mycotoxins, emerging mycotoxins, as well as other fungal metabolites, respectively. Based on the findings of these two studies, it is evident that weather conditions during maize growing seasons strongly influenced the presence of certain fungal metabolites. For example, the rainy and wet conditions in 2014 favored the growth and development of *Fusarium* species and synthesis of *Fusarium* metabolites, while *Aspergillus* metabolites were scarce in maize samples from the same year. Conversely, the hot and dry conditions in 2012, 2013, and 2015 increased the prevalence of *Aspergillus* metabolites, particularly in maize grown under extreme drought in 2012. *Penicillium* metabolites as well as some other fungal metabolites showed consistently high levels across all four years, despite varying weather conditions. It is crucial to highlight that maize samples collected over four years in Serbia encompass a combination of diverse fungal metabolites. This is concerning because co-occurring metabolites have the potential to exhibit synergistic and/or additive effects, which could pose risks to both human and animal health.

The continuation of these studies was carried out in the following years, applying a multi-toxin method [36] for analyzing the presence of *Aspergillus* [24], *Fusarium* [38], and *Penicillium* [23] metabolites in maize samples from 2016 and 2017. In 2016 and 2017, there were no recorded extreme weather conditions, while 2017 was characterized by higher air temperatures and lower precipitation compared to 2016. The increased precipitation during 2016 facilitated the development of *Fusarium* mold, while the higher air temperatures and drier weather during 2017 favored the development of *Aspergillus* mold. Therefore, *Fusarium* metabolites predominated in maize samples in 2016, whereas *Aspergillus* metabolites were more prevalent in maize in 2017. Among the 32 investigated *Aspergillus* metabolites, 9 and 20 were detected in maize samples collected in 2016 and 2017, respectively. Kojic acid and 3-nitropropionic acid were the most frequently detected metabolites in maize samples in 2017, while AFB1 was detected in 21% of maize samples. 30 *Fusarium* metabolites were detected in maize samples in 2016, while 34 were found in samples in 2017. Out of a total of 45 tested *Penicillium* metabolites, 16 were detected in samples in 2016, while 18 were found in samples in 2017. The most prevalent metabolites were oxaline, which was detected in more than 90% of the analyzed samples, as well as questiomycin A, 7-hydroxypestalotin, pestalotin, and mycophenolic acid. Despite significant variations in weather conditions from 2012 to 2017, certain fungal metabolites were consistently detected at exceptionally high frequencies throughout these years. Among the *Fusarium* metabolites, FUMs, particularly FB1, along with MON, bikaverin, siccanol, and BEA, were prevalent. Additionally, notable among the *Penicillium* metabolites were questiomycin A, oxaline, 7-hydroxypestalotin, and pestalotin [22,23,24,36,38].

Due to the established high presence of moniliformin in maize from Serbia, and considering that it is one of the most commonly found fungal metabolites in maize globally [64,65], Radić et al. [42] continued to monitor its presence in maize from Serbia in the following years. The study examined the presence of moniliformin in maize samples from Serbia collected between 2018 and 2022. The results showed that all samples were contaminated with MON, with varying concentrations, particularly across different years. The highest MON levels were recorded in maize harvested during the dry and hot conditions of 2021. In contrast, MON concentrations in maize from 2018 to 2020 were about two to three times lower than those in 2021. The results from the period 2018–2021 corroborate earlier findings from the periods 2012–2015 [36] and 2016–2017 [38], indicating that moniliformin was consistently present in a high percentage (86–100%) in Serbian maize throughout the 11-year study duration.

During the period 2018–2021, the same group of authors continued to monitor the presence of aflatoxins [9] and *Fusarium* toxins [10] in maize from Serbia. The study published by Pleadin et al. [9] once again confirmed that weather conditions during maize cultivation are an extremely important factor contributing to the level of aflatoxin contamination in maize. Hot and dry weather in 2021 resulted in the presence of aflatoxins in as much as 84% of the examined maize samples. In contrast, aflatoxins were detected in fewer than 10% of samples in the years 2018–2020. The same group of authors conducted a comparative analysis of aflatoxin occurrence in maize from Serbia spanning from 2009 to 2021. They found that aflatoxins were present in 72%, 24%, 37%, 31%, and 84% of maize samples from Serbia in the years 2012, 2013, 2015, 2017, and 2021, respectively. In contrast, aflatoxins were completely absent in the years 2009–2011 and 2014 and were present in less than 10% of samples in the years 2016, 2018, and 2020. Based on the multi-year results, the authors claimed that Serbia faces a significant and concerning challenge regarding maize contamination with aflatoxins. They suggested that the increased occurrence of aflatoxins in certain years may be associated with drought conditions and elevated air temperatures during maize cultivation in the summer months.

The same group of authors continued their investigation and analyzed the presence of *Fusarium* mycotoxins, FB1, FB2, ZEN, T-2, and HT-2 toxins, in maize samples collected in Serbia during the period 2018–2021 [10]. The investigated mycotoxins were detected in the following ranges of percentages: DON in 5–13%, ZEN in 0–2%, T-2 in 12–66%, HT-2 in 6–12%, FB1 in 89–100%, and FB2 in 88–100%. Of the six investigated *Fusarium* mycotoxins, FB1 and FB2 were the most frequently present and simultaneously detected at the highest concentrations. Among samples collected over different years, both FB1 and FB2 were found at the highest concentrations in samples from the year 2021. The fact that FUMs were detected in the highest concentrations during 2021 may be attributed to the warm and dry weather recorded during the summer months, which were favorable for the development of certain *Fusarium* molds and the synthesis of toxins.

Based on the published findings regarding mycotoxins occurrence in maize from Serbia, it is evident that maize contamination is strongly associated with weather conditions during the maize growing season, especially those prevailing in the summer months. More frequent occurrences of insufficient precipitation, extreme temperatures, and drought conditions have affected maize growth, significantly stressing maize plants and increasing the risk of poor kernel development. On the one hand, weather conditions during the 15-year study period occasionally led to high levels of mycotoxins, primarily aflatoxins and fumonisins, in maize. These concentrations often exceeded MLs, rendering these samples unsuitable for further use. On the other hand, hot and dry weather conditions also contributed to a reduction in maize production and yields. The years 2012, 2013, 2015, 2017, and 2021, characterized by their hottest and driest conditions, where the highest levels of aflatoxins were detected, along with a significant number of samples not meeting regulatory standards, also experienced the lowest maize yields, significantly below average values. Based on the aforementioned points, it is evident that maize from Serbia frequently faces contamination by mycotoxins, with the extent of contamination largely influenced by weather conditions when irrigation levels are inadequate. Considering that maize is one of the most significant agricultural commodities in Serbia, it is necessary to increase levels of education, investment, and improvement to address this important issue in the future.

#### 3.1.2. Mycotoxins in Maize from Croatia

In Croatia, as in Serbia, there has been a trend of increased maize contamination with mycotoxins over the investigated period of 15 years. *Fusarium* toxins were the most frequently occurring in maize from Croatia, with occasional instances of aflatoxins.

Pleadin et al. [15] examined the presence of DON, FUMs, ZEN, and T-2 toxins using the ELISA method in maize samples collected in 2011 from different regions in Croatia. DON, FUMs, ZEN, and T-2 were detected in 71%, 78%, 90%, and 57% of maize samples, respectively. None of the detected concentrations exceeded the defined MLs. The authors observed that the relatively low concentrations of detected mycotoxins could be associated with very warm and dry weather in the summer months, which is not favorable for many *Fusarium* species. Štefanec et al. [66] investigated changes in DON content in the chain of production, cleaning, drying, storage, and processing of maize. During seven months of storage, the content of DON significantly decreased in comparison to the content detected in the same lots immediately after harvest.

Further, Janić Hajnal et al. [10] investigated the presence of DON, ZEN, FUMs, and T-2/HT-2 toxins in maize samples collected in Croatia over a four-year period from 2018 to 2021. They also summarized published as well as some unpublished data regarding the presence of the listed mycotoxins in maize from Croatia for the period 2012–2017. During the study period, the highest prevalence of *Fusarium* toxins was observed in maize samples in 2014. DON, ZEN, and FUMs were detected in over 90% of the samples examined, while T-2/HT-2 toxins were present in over 80% of the samples. DON and ZEN were detected in concentration ranges of 428–16,350 µg/kg and 15–2596 µg/kg, respectively. In 84% of samples, DON concentrations exceeded the ML of 1750 µg/kg for maize in food, while for ZEN, this was the case in 51% of samples. Concentrations of DON and ZEN surpassing the ML for maize intended as feed material were observed in less than 10% of samples for both mycotoxins. Despite the high occurrence of *Fusarium* mycotoxins, they were also found in significantly higher concentrations compared to other years. Similar to Serbia, in Croatia, the maize growing season in 2014 was characterized by conditions ranging from increased moisture content to extremely wet. The authors directly correlated the unusually high concentrations and occurrence of *Fusarium* mycotoxins with these extremely wet conditions. During the studied period from 2012 to 2021, the extremely wet weather recorded in 2014 was only repeated in May 2019, while in other years, there were occasional instances of slightly to moderately increased precipitation. During this period, except for the year 2014, the most prevalent mycotoxins were FUMs, which were detected in more than 80% of maize samples each year, followed by DON at 10–94%, T-2/HT-2 toxins at 9–84%, and ZEN at 10–63%. Based on these findings, it is evident that *Fusarium* mycotoxins were frequently detected in maize from Croatia over the ten-year period (2012–2021). Variations in weather patterns, particularly increased precipitation during certain months, influencing the higher occurrence of specific *Fusarium* mycotoxins, notably DON and ZEN. In contrast, FUMs were consistently present in maize samples from each of the ten years examined. In years with a higher frequency of *Fusarium* toxin occurrence, there were also higher concentrations observed, resulting in a greater number of samples in which mycotoxins concentrations exceeded the MLs prescribed by regulations [15,67,68].

In several additional studies, the occurrence of certain *Fusarium* mycotoxins in maize from Croatia was examined. Kiš et al. [69] investigated the occurrence of T-2 and HT-2 toxins in maize samples from three Croatian regions during 2017–2018. The results showed that 27% of maize samples were contaminated. The sum of T-2/HT-2 concentrations in two maize samples were higher than the ML recommended by the European Commission (332.3 µg/kg and 252.8 µg/kg, respectively). The study also revealed the relationship between the average precipitation level and high concentrations of the T-2/HT-2 toxin, in which the highest average precipitation was found in the region with the highest concentrations of T-2/HT-2 toxins, but the same was not the case for the environmental temperature. Previous Croatian research on *Fusarium* mycotoxins, that is, T-2 toxin and FUM contamination of maize samples harvested in 2011 from different regions of Croatia pointed to maize contamination with these mycotoxins after the year with extremely high rainfall seen during 2010. The highest concentrations of T-2 toxin and FUMs were 210 and 25,200 µg/kg, respectively [70]. Another study by the same authors, Pleadin et al. [71], investigated the incidence of DON and ZEN in maize harvested in the same year, 2010, in Croatia, following a growth period characterized by extremely high rainfall and low temperatures. Maize samples were collected during the harvest in October 2010, with DON detected in 85% of the samples, reaching a maximum concentration of 17,920 µg/kg, while ZEN was detected in 88% of the samples, with a maximum concentration of 5110 µg/kg. The detection of high concentrations of DON and ZEN once again could be explained by the high humidity and significantly low temperatures in the period of maize growth that might have led to increased contamination of maize with *Fusarium* mold. Another study on T-2 and HT-2 toxin incidence in maize samples sampled from May 2015 to April 2017 showed contamination of 32% of maize samples. The authors reported the highest T-2 toxin concentration of 128 µg/kg and the highest HT-2 toxin concentration of 256 µg/kg, both detected in maize samples compared to other cereals investigated [72].

Pleadin et al. [73,74] investigated OTA and CIT co-occurrence in maize since some *Penicillium* species, such as *P. verrucosum* or *P. citrinum*, produce both OTA and CIT. The results revealed a moderately positive correlation between OTA and CIT concentrations in maize (r_s_ = 0.44) sampled from 2014 to 2016. Out of 88 maize samples, there were 38% CIT-positive and only 6% OTA-positive samples with maximum concentrations of 312 µg/kg and 6.0 µg/kg, respectively. There was a difference in CIT concentration and occurrence according to sampling years, indicating the influence of weather conditions between years. There is also a study on CIT in 158 samples of corn during a five-year period (2017–2021). CIT was detected in 25% of maize samples collected throughout the Croatian region. The highest average concentration was determined in 2016 (162.9 ± 162.0 μg/kg) and 2020 (154.9 ± 358.8 μg/kg), and the highest concentration of 968.6 μg/kg was determined in 2020. The results indicate a wide variation in the determined concentration of CIT, that is, to this sporadic occurrence with high concentrations in individual samples, which can be connected to climatic and other conditions during maize cultivation, that is, inappropriate storage conditions [73,74].

In terms of aflatoxins, the occurrence of AFB1 in maize has been most frequently studied [9,19,61]. During the investigated period, starting from 2009, AFB1 was not detected in maize samples from Croatia collected in 2009–2011 or 2014. In samples from 2016, 2018, 2019, and 2020, AFB1 was detected in less than or around 10% of samples. Approximately 30% and 40% of samples from 2015 and 2017, respectively, were contaminated with AFB1. In contrast, more than 70% of samples from 2012 and 2021 contained AFB1. Pleadin et al. [9] analyzed the presence of AFB1 in maize samples collected from various regions of Croatia over a four-year period, from 2018 to 2021. AFB1 was consistently present in 14%, 16%, 19%, and 40% of samples from 2018, 2019, 2020, and 2021, respectively. In 2021, alongside the highest frequency of occurrence, AFB1 was also present in the highest mean concentration (34.1 ± 103.2 µg/kg) and concentration range (1.5–422.2 µg/kg). Although deviations in average air temperature were frequently recorded during the summer months over the period 2018–2021, drought conditions were predominantly observed in 2021. June and September in 2021 were characterized by minor drought, while July was characterized by severe drought. In the other months of the investigated years, minor drought was only recorded in September 2018, as well as in May 2020. The authors attributed the increased frequency of AFB1 occurrence in maize samples from 2021, compared to the period of 2018–2020, to differences in weather conditions. AFB1 was also previously investigated over a 5-year period from 2009 to 2013. Out of 972 maize samples, AFB1 was detected in 31%, whereas in 22% of samples, AFB1 was detected in concentrations over the ML. The results revealed the AFB1 occurrence to be significantly dependent on the cultivation region, with the highest levels generally found in maize harvested in 2013 that could most likely be associated with weather conditions due to extreme warm and drought weather observed in 2012 during the maize growth and harvesting periods [19,60].

In the investigated period of 15 years, studies of mycotoxins in maize were mostly performed with the ELISA method, later using LC-MS/MS method as confirmation for positive samples. Future investigation should include multi-mycotoxin LC-MS/MS analysis to obtain data on all mycotoxins that can contaminate maize as well as other cereals together, allowing better insight into their contamination and factors such as weather conditions that affect it.

### 3.2. Other Cereals

In Croatia and Serbia, wheat is the most important staple food, while other cereals such as oats, rye, barley, and their derivatives are becoming increasingly important in the human diet. Over the last 15 years, the average production of wheat and barley in Serbia was 2.6 Mt (ranging from 1.6 to 3.6 Mt) and 0.4 million tons (ranging from 0.2 to 0.6 Mt), respectively. Furthermore, in Croatia, during the year 2023, wheat and barley were produced at 161.8 and 321.9 thousand ha, respectively [59].

According to the literature, these cereals are highly susceptible to the development of different toxigenic molds, as well as contamination by various mycotoxins. From Serbia, there are only a few publications regarding the occurrence of mycotoxins in cereals during the investigated period from 2009 to 2023, while a larger number of publications have been published in Croatia. However, it is important to note that, unlike studies on mycotoxin occurrence in maize, there is a lack of systematic research tracking mycotoxins occurrence in other cereals over multiple years.

#### 3.2.1. Mycotoxins in Cereals from Serbia

In Table 1, available results from Serbia regarding the occurrence of mycotoxins in cereals, excluding maize, are presented.

Jajić et al. [75] analyzed the presence of DON in wheat and barley samples from 2010. They reported that 78% of wheat and 27% of barley samples had DON concentrations ranging from 64 to 4808 µg/kg and 118 to 355 µg/kg, respectively. The contamination level exceeded the established MLs in 13% of the samples, as outlined by both the European Commission and Serbian regulations. The authors attributed this contamination to the weather conditions during the spring and summer of 2010, particularly noting significantly higher rainfall compared to long-term average values, which strongly influenced DON production. In the same year, 2010, Jakšić et al. [61] investigated the presence of DON and FUMs in wheat samples. The authors reported that 51% of wheat samples were contaminated with DON, while 65% contained FUMs. FUMs were not detected at higher concentrations, while the authors associated the high concentrations of DON detected in several samples with unusual weather conditions characterized by increased moisture.

In addition to the studies published by Jakšić et al. [61] and Jajić et al. [75], there is only one more study that investigates the presence of mycotoxins in wheat samples. Janić Hajnal et al. [76] analyzed the presence of *Alternaria* toxins, including AOH, AME, and TeA in wheat samples from Serbia collected between 2011 and 2013. *Alternaria* toxins were detected in 78% of the samples, with TeA being the most prevalent (68%), while AOH was present in 12% and AME in 6% of the samples. The co-occurrence of two or three *Alternaria* toxins in wheat samples rarely occurred. In this initial preliminary report on the natural presence of *Alternaria* toxins in wheat from Serbia, the authors reported that humid conditions and higher temperatures during the period from flowering to wheat harvesting were necessary for the occurrence of AOH and AME. On the other hand, TeA is more often present, and weather conditions influence its frequency and concentration levels.

Despite several studies on the occurrence of mycotoxins in wheat, there is a lack of research from Serbia examining the presence of mycotoxins in other cereals, such as oats, rye, and barley.

#### 3.2.2. Mycotoxins in Cereals from Croatia

Table 2 summarizes the results published in Croatia from 2009 to 2023 regarding the occurrence of mycotoxins in cereals.

At the end of 2011, Pleadin et al. [15] examined the presence of DON, ZEN, FUMs, and T-2 toxin in wheat, barley, and oat samples collected from six Croatian regions. In more than 50% of wheat samples, both DON and ZEN were detected. Other tested mycotoxins were found in a lower percentage in the analyzed cereals. None of the concentrations of *Fusarium* mycotoxins exceeded the regulatory ML. In another study by Pleadin et al. [68], T-2/HT-2 toxins were investigated in wheat, oat, and barley samples collected from October 2014 to June 2015. Barley, wheat, and oats showed contamination frequencies of 30%, 32%, and 69%, respectively. The highest concentrations, ranging from 11.6 to 304.2 µg/kg, were found in oat samples. The authors attributed increased contamination of cereals with T-2/HT-2 toxins in 2014 to higher air temperatures and increased humidity during the cereal growth period, as well as extreme humidity during harvesting. The same authors [77] extended their research during the harvest period from July to October 2015, analyzing wheat, oats, barley, and rye for AFB1, OTA, DON, ZEN, and FUMs. AFB1 and OTA were the least frequently detected mycotoxins, with OTA present in more than 20% of barley and rye samples. Conversely, DON was found in over 50% of all cereals tested. Moreover, ZEN was detected in more than 50% of wheat and barley samples, while FUMs occurred in about 40% of the cereals. The study also found no significant difference in mycotoxins contamination between organic and conventionally grown cereals.

In 2016 and 2017, Kovač et al. [78] investigated 11 regulated mycotoxins (AFB1, AFB2, AFG1, AFG2, DON, FB1, FB2, ZEA, T-2, HT-2, and OTA) in wheat, barley, rye, and oat samples from Croatia. *Fusarium* mycotoxins, particularly DON, were the primary contaminants of wheat, detected in 72% of samples in 2016 and 33% in 2017. Additionally, HT-2 toxin and OTA were found in wheat samples from 2016. Favorable moderate temperatures and high humidity during plant flowering in 2016 promoted *Fusarium* species growth and DON synthesis in wheat. Variances in weather conditions contributed to higher wheat contamination levels in 2016 than in 2017. Notably, all detected mycotoxin concentrations in wheat were below prescribed MLs, with only a few samples containing multiple mycotoxins. Barley samples from 2016 showed no contamination, and in 2017, only one sample contained HT-2. However, several rye and oat samples from 2017 exhibited contamination with ZEN, T-2, and HT-2.

The LC-MS/MS method developed by Sulyok et al. [13] was used to analyze mycotoxins and fungal metabolites in various cereal varieties in 2016 and 2017 [81] as well as in different winter wheat varieties in 2019 [37]. Among 117 mycotoxins and fungal metabolites tested, Kifer et al. [81] reported the presence of 17 regulated mycotoxins and their derivatives, 14 ergot alkaloids, as well as 23 *Fusarium*, 18 *Aspergillus*, 18 *Penicillium*, 7 *Alternaria*, and 20 unspecified metabolites. Additionally, in four different winter wheat varieties in 2019, a total of 36 fungal metabolites were quantified [37]. Among these, the most abundant mycotoxins identified were DON, culmorin, 15-hydroxyculmorin, 5-hydroxyculmorin, and aurofusarin.

Kovač et al. [79] investigated the presence of 11 regulated mycotoxins in wheat samples from Croatia. The only detected mycotoxin was DON, presented in 68% of the samples. All detected concentrations of DON were within the MLs for mycotoxins in foodstuffs specified by the Commission Regulation [44]. Furthermore, other examined mycotoxins were not detected in any of the samples. In the same year, Pleadin et al. [80] examined the occurrence of rarely studied ergot alkaloids in wheat and rye samples. The results obtained in this study indicate that a low level of ergot alkaloids contaminated 2% and 18% of wheat and rye samples, respectively. Given the rarity of studies on the occurrence of ergot alkaloids in cereals and the variability of these mycotoxin levels influenced by various factors, the authors indicate that further research over an extended period is necessary to draw conclusive findings.

### 3.3. Milk and Dairy Products

Milk has important nutritional properties and plays an essential role in the human diet across various age groups. As one of the main components of the human diet, milk and dairy products are particularly crucial for infants and young children, as they provide essential nutrients for their growth and development [82]. In Serbia and Croatia, cow’s milk and dairy products derived from cow’s milk are most commonly used, while milk and dairy products from goats, sheep, and donkeys are less prevalent in the human diet. Furthermore, for these two countries, the milk and dairy industry represents economic importance, contributing significantly to agricultural production, trade, and overall economic growth. However, over the past decade, there have been changes in the milk market in Croatia and Serbia, primarily involving a reduction in livestock numbers and a decrease in the number of dairies, particularly those with small-scale production, alongside an increasing reliance on milk imports [83,84]. Furthermore, in recent years, starting mainly from 2012, the presence of AFM1 in milk has become increasingly common, posing significant challenges to the milk and dairy products sector in Serbia and Croatia along the entire supply chain. The authors of numerous studies predominantly attribute this phenomenon to climate change, which has primarily resulted in increased and more frequent contamination of maize with AFB1, subsequently transferred to the organisms of dairy animals through animal feed and transformed into AFM1 [17,85,86,87]. This is particularly concerning because milk, as a primary component of human nutrition, poses the highest demonstrated risk for introducing aflatoxins into the human diet among food items. Moreover, infants and young children consume proportionally more milk relative to their size compared to adults, making them particularly vulnerable to the effects of AFM1 due to their elevated milk consumption. It is necessary to emphasize that the presence of AFM1 in contaminated raw milk poses a significant risk for the occurrence of AFM1 in processed milk and dairy products. This is particularly concerning because AFM1 cannot be destroyed or inactivated by the thermal processing methods commonly used in the dairy industry, such as pasteurization and ultra-high-temperature treatment. Furthermore, contaminated milk poses a significant risk for further contamination of the food chain, as milk is used as a raw material in the production of many food products [88,89].

Considering the numerous adverse effects that AFM1 can have on human health, Kuiper-Goodman [90] assessed the tolerable daily intake (TDI) for AFM1 to be 0.2 ng/kg body weight, while a calculated hazard index (HI) higher than 1 indicates a risk to consumers. Furthermore, numerous countries worldwide have established regulatory limits for AFM1 in milk. The most stringent standard, set by the EU [44], mandates an ML of AFM1 in milk at 0.05 µg/kg. Croatia, being a member of the EU, adheres to this standard by setting its ML at 0.05 µg/kg. Serbia aligned its regulation regarding AFM1 with those of the EU in 2011. However, owing to the increased and more frequent occurrences of AFM1 in milk, the initial ML was revised from 0.05 to 0.50 µg/kg. Subsequently, this ML has undergone multiple revisions. Over a decade since the initial amendment, the maximum allowable level currently remains at 0.25 µg/kg, which still does not align with EU regulations [47].

From 2012 to date, a considerable number of studies have been published on the occurrence of AFM1 in milk in both Serbia and Croatia. It is assumed that the high level of milk contamination in both countries at the end of 2012 and beginning of 2013 is the reason for the large number of studies. Most of the studies from Croatia have been published based on the results obtained in one of the reference laboratories for determining AFM1 in milk, namely the Croatian Veterinary Institute, while published studies from Serbia originated from various laboratories, faculties, and institutes. In this study, a review of literature from Serbia and Croatia regarding the occurrence of AFM1 in milk and dairy products in Serbia and Croatia was conducted.

#### 3.3.1. AFM1 in Milk from Serbia

In the observed period, from 2009 to 2023, the initial notable presence of AFM1 in milk from Serbia was observed at the end of 2012 and the beginning of 2013. Kos et al. [17] and Škrbić et al. [18] investigated the presence of AFM1 in milk from Serbia collected during the first half of 2013. They reported that in 86% and 76% of samples, respectively, the AFM1 concentration exceeded the ML (0.05 µg/kg) defined by both the EU and the Serbian Regulations, valid at that time. The authors of these studies indicated that the significant contamination of milk with AFM1 during the 2012–2013 period was a consequence of maize contamination with AFB1. Specifically, as mentioned earlier, the maize growing season in 2012, particularly during the summer months, was characterized by moderate to extreme drought conditions that resulted in aflatoxin presence in more than 70% of maize samples [22,91]. Due to the significant milk contamination levels, regulatory adjustments in Serbia were made in 2013 to modify the ML for AFM1 in milk, raising it from 0.05 to 0.50 µg/kg. In their studies, Škrbić et al. [18] and Kos et al. [17] investigated the estimated daily intake of AFM1. Škrbić et al. (2014a) utilized the average level of AFM1 in milk samples and the milk consumption rate by the average Serbian adult consumers for estimation. They found that the estimated intakes of AFM1 varied across different months, with values of 1.420, 0.769, and 0.503 ng/kg bw/day for February, April, and May in 2013, respectively. The analysis revealed that the calculated hazard index for February, April, and May was almost 7, 4, and 3 times higher than 1, suggesting a serious risk of AFM1 to Serbian consumers. Furthermore, Kos et al. [17] based their estimation on data collected from 1500 respondents, covering 5 different age categories, regarding their body weight and milk intake. In this study, an estimated daily intake of AFM1 was conducted for obtained maximum, minimum, and average concentrations of AFM1. The results revealed significantly higher values than 0.2 ng/kg body weight defined by Kuiper-Goodman [90] for maximum and mean AFM1 concentrations, whereas for minimum concentrations, the estimated intake of AFM1 was higher only for children aged 1 to 5 years. Overall, the authors concluded that children were at the highest risk of AFM1 exposure due to their increased milk consumption and lower body weight. Due to the previously mentioned elevated contamination of maize with aflatoxins and a high percentage of non-compliant AFM1 concentrations in milk compared to current regulations, Serbia experienced the so-called “aflatoxin crisis” in early 2013. This crisis prompted the replacement of the Minister of Agriculture, triggered numerous protests by agricultural workers, led to several amendments to the Regulation for MLs of AFM1 and AFB1, caused confusion among consumers due to contradictory media information, garnered exceptional media attention, and resulted in a decrease in the purchase of milk and dairy products. Furthermore, this situation incurred great economic losses.

After the initial escalated milk contamination with AFM1 at the end of 2012 and the first half of 2013, further instances of milk contamination occurred in subsequent years. Tomašević et al. [92] investigated the presence of AFM1 in milk and dairy products in the period 2013–2014, Miocinovic et al. [93] focused on milk in 2015, Milićević et al. [86] examined samples from 2015 to 2018, and Udovicki et al. [94] analyzed data spanning from 2015 to 2022.

Tomašević et al. [92] reported that among 1438 samples, 56% of raw milk, 33% of heat-treated milk, and 38% of milk samples had AFM1 concentrations exceeding the ML set by the EU Regulation. However, the authors indicated differences in contamination levels among seasons. In winter and spring, AFM1 mean concentrations (0.358 µg/kg and 0.375 µg/kg, respectively) did not significantly differ, while concentrations in the summer (0.039 µg/kg) and autumn (0.103 µg/kg) were significantly lower. Additionally, the percentage of non-compliant samples exceeding the ML decreased from 62% to 12% by the end of 2014. The noted differences in the level of milk contamination with AFM1 can also be attributed to maize contamination with aflatoxins. Specifically, Kos et al. [91] reported that maize from 2012, 2013, and 2014 was contaminated with aflatoxins at rates of 72%, 25%, and 0%, respectively. Such significant differences in contamination largely contributed to variations in the level of milk contamination with AFM1, as well as to the decreased milk contamination by the end of 2014.

Miocinovic et al. [93] conducted a study on the occurrence of AFM1 in milk and dairy products throughout 2015. Their findings revealed elevated AFM1 levels in raw milk during the latter half of the year, with 50% of milk samples failing to meet EU standards. They proposed that the increased incidence of AFM1 in the latter part of the year could be attributed to the consumption of contaminated maize in the feed of lactating animals. This finding is consistent with the findings reported by Kos et al. (2018), which indicated that out of 600 maize samples collected in 2014 and 2015, none tested positive for AFB1 in 2014, whereas 37% of samples in 2015 were contaminated with AFB1. Furthermore, a more sensitive analytical method (LC-MS/MS) applied to the analysis of maize samples in 2015 revealed that up to 90% of them were contaminated by aflatoxins. As a consequence, it can be inferred that the maize harvested in 2014 primarily served as feed for cows during the initial six months of 2015, whereas the prevalence of maize from 2015 surged in the latter half of the year, resulting in fluctuations in milk contamination levels throughout the year.

One of the largest studies from Serbia regarding the occurrence of AFM1 in milk was conducted by Milićević et al. [86]. This study was based on examining the presence of AFM1 in more than 20,000 milk samples collected over a period of four years, from 2015 to 2018. The results revealed that 70%, 85%, 79%, and 82% of milk samples collected in 2015, 2016, 2017, and 2018, respectively, were found to be contaminated. Moreover, in 30%, 31%, 27%, and 22% of milk samples from the corresponding years, the concentrations of AFM1 surpassed the ML of 0.05 µg/kg set by the EU. Milićević et al. [86] noted that the prevalence of AFM1 exhibited periodic fluctuations throughout the survey period. Despite the absence of aflatoxin contamination in maize samples in 2014, a high occurrence of AFM1 in raw milk (70%) was observed in 2015. The authors suggested that milk contamination may have resulted from the use of cattle feedstuffs or feed with elevated levels of AFB1 due to inadequate storage conditions. In contrast, the AFM1 presence in milk in other surveyed years may be attributed to the occurrence of AFB1 in maize samples.

Another extensive study, which included about 14,000 samples from 2015–2022, was published by Udovicki et al. [94]. They found that 78% of milk and between 42% and 79% of various dairy products were contaminated with AFM1. Furthermore, in 14% of milk samples, the AFM1 concentration exceeded 0.05 µg/kg. In the same study, the authors evaluated chronic dietary exposure to AFM1 in Serbia. They collected information on milk and dairy product intake using a Food Frequency Questionnaire validated by a 24-h recall-based method. Risk characterization was performed by calculating the margin of exposure as well as the number of hepatocellular carcinoma cases induced by AFM1. The authors found that the mean estimated daily intake of AFM1 was highest in children, followed by adolescents, adult females, and adult males. The margin of exposure values based on the mean estimated daily intake for all population groups were above the risk-associated threshold, and the number of possible hepatocellular carcinoma cases ranged from 0.0002 to 0.0021 cases per year for every 105 individuals. The results obtained in this study suggest low health risks due to AFM1 exposure for the entire population, but the authors highlighted that the risk is not non-existent, particularly for children, as they have a higher proportion of the population exposed to AFM1 levels associated with risk.

Apart from the aforementioned investigation, several other investigations conducted in Serbia have evaluated the risk linked to the presence of AFM1 in milk and dairy products [17,18,20,21,94,95]. These studies employed diverse methodologies for risk assessment and incorporated various data on milk consumption, resulting in a range of estimations regarding daily intake. Nevertheless, a consistent finding across all studies is the elevated risk for children posed by AFM1 in milk, given their proportionately higher milk intake relative to body weight. Furthermore, the years with the most pronounced maize contamination coincided with increased AFM1 concentrations in milk as well as the highest estimated daily intake.

#### 3.3.2. AFM1 in Milk from Croatia

Similar to Serbia, increased contamination of milk with AFM1 was noticed in Croatia in 2013. Between February and July 2013, over 4000 samples from Croatia, including both raw and ultra-high temperature (UHT) milk, were analyzed to investigate the presence of AFM1 [96]. The results revealed significant statistical differences in mean AFM1 concentrations between raw and UHT milk samples collected during specific months. Notably, the highest AFM1 concentration observed in February led to an excess of ML in 46% of raw and 36% of UHT milk samples. Overall, AFM1 levels surpassed ML in 28% and 10% of raw and UHT milk samples, respectively. The authors attributed the higher AFM1 concentrations during winter compared to spring and summer months, particularly in the May–July period, to the use of more fresh raw feed instead of concentrated feed during the summer months. The authors reported that the high AFM1 concentrations, as well as the high percentage of samples exceeding the ML, were attributed to the weather conditions in Croatia during 2012. These conditions, as stated earlier, were characterized by warm weather and a lack of rain, leading to prolonged drought, especially during the maize growth and harvesting period.

After the increased occurrence of AFM1 in milk at the end of 2012 and the first half of 2013, a group of authors Bilandžić et al. [85] continued to monitor the presence of AFM1 in milk from Croatia. More than 3000 milk samples were collected from October 2013 to September 2014. The results indicate a significantly lower level of contamination compared to the previous year. The authors attribute this lower contamination level to better weather conditions during the cultivation of cereals used for cattle feed, primarily maize grown in the year 2013. Also, the authors indicated that the outbreak of the crisis due to elevated AFM1 levels in the first half of 2013 resulted in a more careful approach to the control of supplementary feedstuff for lactating cows. Only during the period of October–December 2013, AFM1 concentrations exceeding the ML were found in less than 10% of the samples, while in other periods, concentrations rarely or none at all exceeded the ML. Furthermore, the authors reported that elevated AFM1 concentrations were detected in only a few samples, attributing this occurrence to sporadic and localized usage of contaminated feed for dairy cows, particularly on small farms. During the same year, from July to September 2013, Bilandžić et al. [97] investigated the presence of AFM1 in raw cow, goat, sheep, and donkey milk. The results indicated that in cow milk, 7% of samples exceeded the ML for AFM1 concentration, while other cow milk samples, as well as the other types of milk investigated, were either free of AFM1 or contaminated with AFM1 at concentrations lower than 0.05 µg/kg.

The same group of authors [98] continued their research on the occurrence of AFM1 in milk from Croatia. In a study published in 2017, samples of cow, goat, and sheep milk were investigated. The results of the analysis of milk samples collected during the spring and autumn of 2016 indicated low levels of AFM1 detected in the milk of three dairy species. In this study, the authors also assessed the probable daily intakes of AFM1, and the results showed no risk for consumers in 2016. The decreased occurrence of AFM1 in the milk of various species was attributed by the authors to unfavorable weather conditions for AFB1 synthesis in cereals. Additionally, they noted that farms have been increasingly implementing systematic control measures for AFB1 in animal feed, alongside improvements in production practices and the maintenance of appropriate storage conditions.

Bilandžić et al. [87] further investigated the seasonal occurrence of AFM1 in raw milk over a five-year period, between winter 2016 and winter 2022, in Croatia. Among a total of 5817 samples, 95% had AFM1 concentrations lower than LOQ, while 2% AFM1 concentrations exceeded the ML. Furthermore, in the same study, the authors conducted a dietary exposure and risk assessment for the adult Croatian population. The estimated daily intakes for positive samples ranged between 0.17 and 2.82 ng/kg body weight/day, indicating a high level of concern during autumn and winter, particularly for consumers of large amounts of milk. Additionally, the risk assessment of AFM1 dietary exposure from raw cow milk indicates a high level of concern during autumn and winter from a public health perspective. In conclusion, the authors stated that weather conditions, such as extreme droughts recorded in certain years, contributed to the development of toxigenic molds in maize and the synthesis of AFB1, which influenced the increased frequency of milk contamination with AFM1, especially during the period when dairy cows receive substitute feed in autumn and winter. Considering the influence of the seasons, they found that AFM1 exhibited the highest mean values and statistically significant differences of means between years during the autumn months.

Based on the results from Croatia regarding the occurrence of AFM1 in milk from 2013 to 2023, it is evident that the highest incidence, with the greatest concentrations of AFM1, was recorded in 2013. The authors of the studies associated the high levels of milk contamination with AFM1 during 2013 with the exceptionally dry summer in 2012 and the contamination of maize with AFB1, which was utilized in the feeding of dairy animals, primarily cows, in 2013. Subsequently, in later years, a lower incidence of AFM1 and a significantly lower percentage of samples with AFM1 concentrations above the ML were observed.

### 3.4. Meat and Meat Products

Meat and meat products are important sources of protein in the human diet, and their consumption depends on socioeconomic factors, ethical or religious beliefs, and traditions [99]. In Croatia, poultry meat ranks first in terms of consumption per household member, followed by pork and beef [100]. In Serbia, pork is traditionally the most consumed type of meat, and this also correlates with pork production, which is by far the most commonly produced meat, followed by poultry and beef, which are also highly valued [101]. One of the most popular groups of meat products in many European countries is traditional fermented pork meat products. The main mycotoxins associated with meat and meat products include AFB1, OTA, CIT, cyclopiazonic acid (CPA), and STC [102,103]. Their appearance in this type of food is linked to three possible sources of direct or indirect origin, including: surface molds, which overgrow dry-cured meat products during ripening, the use of contaminated spices into their production, and as a consequence of the carry-over effect from contaminated feed used in the diet of farm animals [104]. However, the MLs of mycotoxins in meat products are not yet established in the EU.

The mycotoxins that generally have the greatest public health significance in terms of their toxicity and occurrence in this food category are AFB1 and OTA. It was found that OTA is the dominant contaminant of meat products, while AFB1 appears less frequently and in much lower concentrations [28,29,32,105,106]. In addition to those most important, some types of molds found on the surface of meat products can also produce mycotoxins CIT, CPA, and STC, but their impact on the occurrence and quality and safety of meat products, and consequently on human health, is still not sufficiently investigated [102,107,108]. The possible co-occurrence of CIT and OTA, AFB1 and STC, and AFB1 and CPA has also been described in the literature [107,108,109]. Some authors have highlighted the potential presence of CPA in meat and meat products, even in very high concentrations [102,110]. Consequently, in recent years, research has focused on the development of sensitive analytical methods for CPA analysis. The occurrence of this mycotoxin in high concentrations in certain types of meat has been confirmed [30,34,111].

Various spices are used in the meat industry to give meat products a characteristic taste, among which red, white, and black pepper, sweet and hot pepper, mustard, garlic, pepper, rosemary, and cinnamon are commonly used all over the world. It is known that spices are mainly imported from developing countries with tropical and/or subtropical climates. In these regions, conditions such as high temperatures, heavy precipitation, and humidity often encourage mold growth and mycotoxins production. This can lead to the contamination of spices with mycotoxins, as a consequence of their contamination by molds of the genus *Penicillium* and *Aspergillus* [103]. Additionally, spices are often dried on the ground outdoors in poor hygienic conditions, further encouraging mold growth and mycotoxin production [112]. High levels of AFB1 (155.7 µg/kg; 75.8 µg/kg) [112] and OTA (177.4 µg/kg; 79.0 µg/kg) were recorded in spices that are most often used in the meat industry, such as paprika and black pepper [26,113]. The first data on STC in spices was related to spices from India, in which STC was detected in black pepper (105–125 μg/kg) and fennel (142 μg/kg) [114]. There was also an investigation on the presence of CIT in spices, such as black pepper, in which it was found in concentrations of up to 50 μg/kg [115].

Furthermore, it is also known that the presence of mycotoxins in meat and meat products can occur as a result of feeding domestic animals with contaminated food during the production process [25,116]. The term *carry-over* refers to the transfer of unwanted compounds from contaminated feed to food of animal origin, in this case, mycotoxins to meat and offal and, finally, to meat products. In farm animals, higher ratios of transfer of some mycotoxins (primarily OTA) to liver and kidney tissue can be seen compared to other organs or muscle tissue. For example, feeding pigs for 40 days with 0.68 mg OTA per day resulted in OTA concentration in smoked hams ranging from 1.26–5.65 µg/kg [117]. Concentrations naturally vary depending on the length of exposure, dose, and method of intake, but the highest levels were recorded in kidney tissues, blood, and blood plasma, which actually represents a danger for the production of specialties such as blood sausage, pate, and sausages, in the production of which the above ingredients are used. AFB1 has also been reported to carry-over into pig meat in liver, muscle, and fat tissue [118]. Recently conducted research for CIT also shows the possibility of a carry-over effect in edible pig tissues but with a low transfer rate (0.1–2%) [119]. However, all the details of mycotoxin transfer from feed to meat and meat products have not yet been fully elucidated, and further research is needed.

Numerous studies have shown that the mycotoxins found in dry-cured meat products with long periods of ripening during their production are largely the result of the production of toxic species of surface molds of the genera *Penicillium* and *Aspergillus* [120,121,122]. However, the growth of toxigenic molds does not necessarily mean that mycotoxins are present in the food, and likewise, the absence of visible molds does not mean that there are no mycotoxins present. The production of mycotoxins in meat products is influenced by various biological and environmental factors [123]. (Surface molds produce mycotoxins under certain conditions of production and storage, namely temperature, pH value, a_w_, damage to the casing, presence of skin, cracks, and insufficient washing and brushing of the dry-cured meat product surface, i.e., uncontrolled mold growth [32]. Production conditions can directly or indirectly affect the formation of mycotoxins or affect the growth and development of molds themselves, the availability of precursors needed in biosynthesis, or the expression and activity of enzymes involved in the biosynthetic pathway [123].

Environmental conditions affect mycotoxin production to a greater extent than mold growth. High temperature and humidity are generally considered to increase the risk of mold growth and mycotoxin production, with warmer and drier weather thereby being better suited for *Aspergillus* and colder and rainier weather better for *Penicillium* species growth [32]. Evidence indicates that European countries with moderate climates are more exposed to molds and mycotoxins due to climate change. The climates in these countries are likely to become warmer, reaching a temperature of 33 °C, a temperature very close to optimal growth for *A. versicolores* (30 °C) and STC production (23–29 °C), for example [124].

Unlike plants, which are directly exposed to weather and subsequent mold growth, dry-cured meat products mature in chambers in which some of the environmental factors can be regulated, although this is regrettably not customary for home-based production. Due to the interactions of various factors that may affect mycotoxin biosynthesis by molds during production, the detected mycotoxin concentrations in meat and meat products are difficult to connect only to the weather conditions, bearing in mind that meat products can be contaminated with mycotoxins in other mentioned pathways in which case connection with weather conditions is more difficult to make.

#### 3.4.1. Mycotoxins in Meat and Meat Products from Serbia

Data on the occurrence of mycotoxins in meat and meat products from Serbia can be considered very scarce and are limited to a small number of studies [104]. In general, OTA in meat and offal has been studied more than other mycotoxins in this country. It has been detected in pig’s blood, kidney, liver muscle, and adipose tissue with rather high levels found in animals suffering from porcine nephropathy [125,126]. Average OTA values in pig’s blood, kidney, and liver were 3.70 ± 23.60 µg/kg, 1.26 ± 5.85 µg/kg, and 0.63 ± 1.87 µg/kg, respectively [125].

Polovinski Horvatovic et al. [127] investigated OTA on the health of 95 pig’s kidneys from Vojvodina, Serbia, and found this mycotoxin in 15% of samples in ranges from 0.10 to 3.97 μg/kg. The authors concluded that the obtained results do not suggest any serious problems with OTA contamination in the pig’s kidneys. Milićević et al. [128] concluded that in temperate countries, like Serbia, OTA is mainly produced by *P. verrucosum* and is associated with contamination of several foodstuffs, including pork meat.

Other data about mycotoxin occurrence in meat and meat products as well as offal originating from Serbia are not available in the literature. In the risk assessment of dietary exposure to AFB1 in Serbia, no meat or meat products were included, but AFB1 was detected in dried ground red paprika, a spice commonly used in meat products, in 2% of samples in concentrations of 0.5–2.88 µg/kg but not exceeding the ML of 5 µg/kg [129].

#### 3.4.2. Mycotoxins in Meat and Meat Products from Croatia

Data from Croatia show that mycotoxins can occur in meat and meat products as a result of direct contamination, as a result of the production of toxic molds, or through the so-called carry-over effect, i.e., by transfer from the meat of domestic animals that have consumed feed or feed mixtures contaminated with mycotoxins, and indirectly through contaminated spices and other raw materials used in their production. Data on the occurrence of mycotoxins AFB1, OTA, STC, CIT, and CPA in samples of Croatian traditional meat products by type of product are presented in Table 3.

In the last 10 years in Croatia, there have been studies on mycotoxins in meat and meat products, mostly on AFB1 and OTA and, more recently, CPA. Research by Lešić et al. [30] included all five mycotoxins important for meat and meat products. ELISA was previously the most commonly used analytical method for mycotoxin analysis in meat. However, recently, the LC-MS/MS method has become more frequently utilized. CPA and OTA were most commonly detected, with CPA in very high concentrations, up to 335.5 µg/kg.

Significant transmission was recorded for the mycotoxin OTA through the carry-over effect in pigs that were exposed to a contaminated diet (250 μg OTA/kg of feed) with the main accumulation of OTA in the kidneys (13.87  ±  1.41 μg/kg), lungs (0.47  ±  1.97 μg/kg), and liver (7.28  ±  1.75 μg/kg). Also, in produced final meat products, dry fermented ones, and cooked ones, average concentrations ranged from 4.51 ± 0.11 µg/kg in smoked ham to 14.02 ± 2.75 μg/kg in black pudding sausages [116,130].

According to meat product contamination by surface toxicogenic molds, the influence of climatic conditions and mycotoxin appearance in the final meat products has been proven. The authors concluded that the production of mycotoxins depends on climatic factors that are characteristic of a certain geographical area and vary on an annual basis. The eastern and the western Croatian regions have similar moderate climates, while the southern Croatian region is often compared with tropical and subtropical regions. Higher temperatures and dryness are preferred by *Aspergillus* species, while lower ambient temperatures are more suitable for *Penicillium* species [30,32]. Therefore, OTA and CPA detected in these moderate climate regions are probably mostly produced by *Penicillium*, such as *P. nordicum* and *P. commune*, rather than *Aspergillus* species [30].

To the best of our knowledge, studies on STC occurrence in meat and meat products have generally not been conducted except in one Croatian study, although there is an EFSA recommendation for collecting data on STC in food and feed using highly sensitive analytical methods (EFSA, 2013). For CPA, there are few studies in the last six years on Spanish and Croatian dry-fermented meat products [30,31,34,111],), while CIT in meat and meat products was investigated in two Croatian studies [30,104].

The interaction of climate changes, such as increased carbon dioxide concentration, temperature rise, and extreme changes in rainy and dry periods, have a significant impact on the growth of molds and the occurrence of mycotoxins [131]. Research has shown that environmental factors, such as the combination of temperature × a_w_ × CO_2_, directly affect the expression of AFB1 biosynthetic genes but have no significant impact on *A. flavus* growth [132]. It has also been concluded that temperature is the most important factor affecting the production of CPA, but that the interactions of different factors such as temperature and a_w_ are also important [133]. The optimal temperature for the production of most mycotoxins is between 20 and 30 °C with an a_w_ above 0.65. However, the authors concluded that the impacts of climate change on the formation of mycotoxins in this food category are difficult to predict due to the complex interplay of various factors that also affect their occurrence in final meat products [131].

### 3.5. Other Food

In general, studies on mycotoxins in food other than the most represented cereals and cereal-based products, milk and milk products, and meat and meat products, include nuts, wine, soda and juices, dried fruit, coffee, spices, and others, with most of them having an ML set by the EU. However, research on food types other than cereals, milk and dairy products, and meat and meat products is limited in both Serbia and Croatia.

#### 3.5.1. Mycotoxins in Other Food from Serbia

Available data in literature related to the mycotoxin occurrence in other food from Serbia in the examined period are shown in Table 4.

Torović et al. [134] conducted a study in Serbia focusing on breakfast cereals, examining the presence of AFB1 and OTA in samples collected in 2012 and 2015. OTA was detected in 21% and 13% of the samples collected in 2012 and 2015, respectively. For those years, Torović et al. [134] found that 4% of the samples from 2012 exceeded the MLs. AFB1 was detected in 11% of the samples from 2015 with low concentration levels. The authors concluded that the exposure of the adult Serbian population to AFB1 and OTA did not raise health concerns. Additionally, an extensive study [129] analyzed 463 samples of various food products (peanuts and peanut-based products, maize milling products and maize-based products, tree nuts, rice, millet, mixed composition products, dried figs, and dried ground red paprika) commonly consumed in the country for AFB1 contamination. Overall, an AFB1 occurrence of 29% was detected, in which muesli with cornflakes, millet, dried figs, and maize flour showed the highest percentages of positive samples. In certain samples, concentrations were found to exceed MLs, while the mean concentrations of AFB1 were generally low. However, peanut butter, maize flour, tortilla chips, and cornflakes had the highest means (7.86, 4.67, 3.43, and 3.10 µg/kg, respectively) and maximum AFB1 concentrations (13.10, 28.15, 5.83, and 8.64 µg/kg, respectively). The study suggested increased health risks from AFB1 intake, especially for children and adolescents, with maize-based products being the most significant contributor to overall exposure.

To the best of our knowledge, among studies on mycotoxins in other food, there is patulin analysis in fruit juices, OTA analysis in wine, OTA and aflatoxins in spices, and different mycotoxins in various nuts and biscuits, cookies, dried fruits, and fruit jams. Until 2017, there were no previous studies carried out in Serbia regarding the occurrence of patulin in food, nor the exposure of the Serbian population to this mycotoxin. The presence of patulin in infant fruit juice, infant purée, and juice for infants and preschool children was analyzed [135]. Patulin was found in 44% of infant juices, 17% of infant purée, and 43% of fruit juices for children. None of the samples exceeded the ML. Furthermore, an extension of the initial study by Torović et al. [136] focused on the presence of patulin in apple and multi-fruit juices for children, adolescents, and adults. Patulin was detected in 51% of all examined juices, where apple juice (74%, 6.4 μg/kg) had a higher percentage of contamination and mean content compared to multi-fruit juice (28%, 2.1 μg/kg). The ML was exceeded in 0.7% of all examined samples. In the third study, Torović et al. [137] analyzed the presence of patulin in commercial bilberry and black chokeberry juices. Only one sample of bilberry juice was found to contain patulin, with a concentration of 3 μg/L, well below the ML of 50 μg/L. The risk assessment of patulin intake through fruit juices for Serbian infants, preschool children, children, adolescents, and adults, as conducted in the three studies, showed no health concerns.

The risk to public health related to the presence of OTA in wine collected from the Fruška Gora mountain in Northern Serbia was assessed based on 113 wines from 2011 to 2016 [138]. The presence of OTA in wine samples per year is presented in Table 4. In total, OTA was detected in 52% of analyzed wine samples, where red wine (64%) had a higher percentage of contamination compared to white (43%) and rose wines (36%). All samples analyzed were within the ML for OTA content, indicating a very low risk to public health from OTA exposure. Furthermore, the presence of aflatoxins and OTA in powdered red paprika and peppers was analyzed, and all analyzed samples were found to be free of contamination [139].

The presence of different mycotoxins (aflatoxins, OTA, FUMs, ZEN, T-2, and HT-2 toxin) in various nuts, including walnut, hazelnut, almond, and peanut were analyzed [140]. The results showed weak contamination of nuts, in which ZEN in the concentration range of 1.20–3.48 µg/kg was detected in only 2 walnut samples. Furthermore, Škrbić et al. [141] analyzed the presence of different mycotoxins (aflatoxins, OTA, DON, ZEN, T-2, and HT-2 toxins) in biscuits with fruit filling, cookies, dried fruit, and fruit jams. Most of the samples complied with regulations, but 8 (21%) biscuits and 3 (75%) fig jam samples exceeded the limits for OTA. The study estimated dietary exposure to mycotoxins for children, adolescents, and adults, finding that exposures were below the predicted risk threshold.

Although the current review suggests that mycotoxin exposure of the Serbian population through various food sources does not raise health concerns apart from the mentioned results in the study by Udovicki et al. [129] for AFB1, a more comprehensive investigation is necessary. This should encompass all relevant food items and consider potential co-exposures with other mycotoxins to conduct a thorough exposure assessment and evaluate potential health effects accurately.

#### 3.5.2. Mycotoxins in Other Food from Croatia

Studies on food types other than cereals, milk, and meat products in Croatia are very scarce. Available data in the literature related to mycotoxin occurrence in other food from Croatia in the examined period are shown in Table 5.

Concerning cereal and ceral-based products, there is a study on bread and other bakery products that were tested for the presence of DON. In 17% of the 111 bakery products analyzed, DON was detected in concentrations up to 400 µg/kg, which did not exceed the ML of 500 µg/kg set for this type of food [142].

Furthermore, in Croatia, to the best of our knowledge, among studies on mycotoxins in other food, there is OTA analysis in wine and patulin in apple soda. Bošnir et al. [143] reported that OTA was detected in 65% of analyzed wine samples, where red wine contained higher OTA concentrations in relation to white ones in concentrations up to 1.696 µg/L, but below ML. The study indicated differences in OTA concentrations between different Croatian geographical regions with different clime.

Regarding apple juice, a significant source of patulin in the human diet that is commonly consumed by young children, there is a scientific opinion regarding its presence during the period from 2014 to 2016 in Croatia. Patulin was detected in 21% out of 122 analyzed samples in concentrations up to 154 µg/kg, with 4% of samples exceeding an ML of 50 µg/kg. There is a difference in detected concentrations according to sampling year. In this scientific assessment regarding the presence of patulin in apple juice, the conclusion was drawn that by monitoring mycotoxins over an extended period, uncertainties regarding the influence of climatic factors on the presence of mycotoxins in monitored food can be partially mitigated [144].

## 4. Conclusions

A comprehensive analysis of studies from 2009 to 2023 shows that mycotoxins are frequently present in agricultural raw materials and food in Serbia and Croatia. The results indicate that in Croatia, *Fusarium* toxins, particularly DON, ZEA, and FUMs, are the most prevalent in maize, exceeding MLs during years with extremely wet conditions. Similarly, in Serbia, maize is frequently contaminated by mycotoxins, with FUMs being the most prevalent but rarely in high concentrations. Aflatoxins are found in high concentrations in drought years. Moreover, AFM1 is present in milk and dairy products in both countries, with the highest concentrations observed in 2013, correlated with elevated AFB1 contamination in maize. Mycotoxins are frequently detected in Croatian meat and meat products, with CPA and OTA being the most commonly found. In Serbia, data on mycotoxins in meat and meat products are limited, with more focus on OTA in meat and offal.

Based on these results, there is a clear need for further research on mycotoxin occurrence and risk assessments in various food categories directly consumed by humans in Croatia and Serbia, including cereal-based foods, milk and dairy products, fruit and vegetable products, meat, and other foods. Advances in analytical methods, particularly multi-toxin LC-MS/MS methods, are necessary to simultaneously investigate the presence of the multiple mycotoxins in different samples. Urgent action is required for improvement, investment, and educational initiatives across all levels of agricultural raw material and food production in both countries, considering the documented climate changes and projected future scenarios and their impact on mycotoxin occurrence.

## Figures and Tables

**Figure 1 foods-13-01391-f001:**
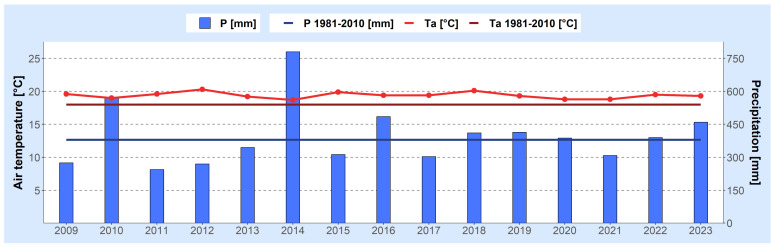
Seasonal (April–September) meteorological conditions in Northern and Central Serbia. The blue bars represent total seasonal precipitation [mm], while the red dots connected by a red line indicate the average seasonal air temperature [°C]. The solid dark blue line shows the average seasonal precipitation, while the solid dark red represents the average seasonal air temperature during the period 1981–2010.

**Figure 2 foods-13-01391-f002:**
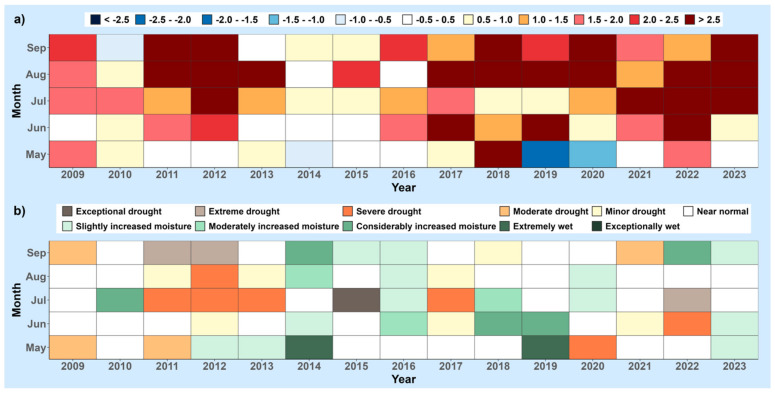
Monthly meteorological conditions in Northern and Central Serbia. (**a**) Monthly air temperature deviations from long-term average 1981–2010; (**b**) Assessment of monthly humidity/dryness using the Standardized Precipitation Index (SPI-2).

**Figure 3 foods-13-01391-f003:**
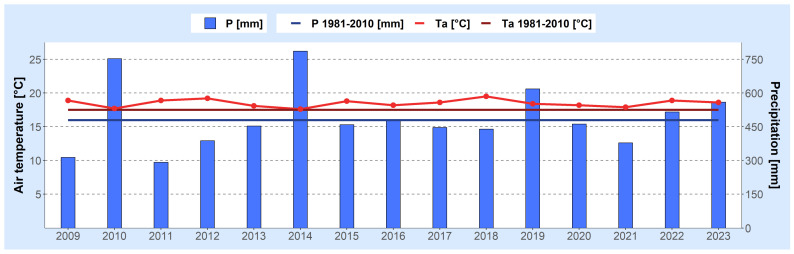
Seasonal (April–September) meteorological conditions in lowland Croatia. The blue bars represent total seasonal precipitation [mm], while the red dots connected by a red line indicate the average seasonal air temperature [°C]; The solid dark blue line shows the average seasonal precipitation, while the solid dark red represents the average seasonal air temperature during the period 1981–2010.

**Figure 4 foods-13-01391-f004:**
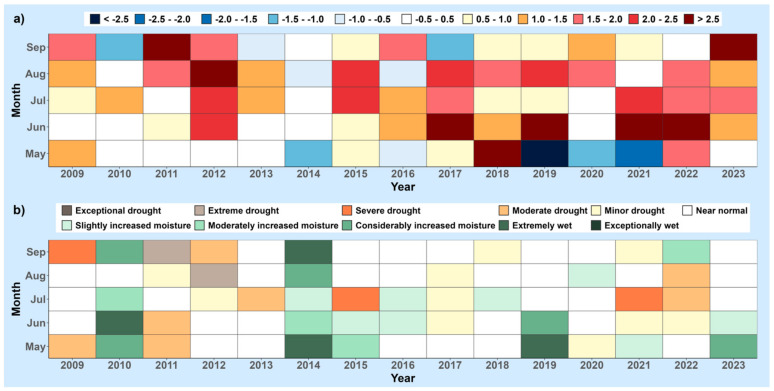
Monthly meteorological conditions in lowland Croatia. (**a**) Monthly air temperature deviations from long-term average 1981–2010; (**b**) Assessment of monthly humidity/dryness using the Standardized Precipitation Index (SPI-2).

**Table 1 foods-13-01391-t001:** Occurrence of mycotoxins in cereals from Serbia.

Type of the Product	Mycotoxin	Sampling Year	No of Samples	% of Positive Samples	Range of Concentration (μg/kg)	Reference
Wheat	DON	2010	128	78	64–4808	[75]
Barley	DON	2010	11	27	118–355	
Wheat	DON	2010	25	51	64–1604	[61]
	FUMs	2010	25	65	27–614	
Wheat	AOH	2011	40	2	1.3	[76]
		2012	39	8	12.2–41.0	
		2013	13	54	0.75–48.9	
	AME	2011	40	-	-	
		2012	39	5	54.1–70.2	
		2013	13	31	0.49–62.3	
	TeA	2011	40	85	2.5–62.2	
		2012	39	56	2.5–115.0	
		2013	13	54	7.5–267.6	

**Table 2 foods-13-01391-t002:** Occurrence of mycotoxins in cereals from Croatia.

Type of the Product	Mycotoxin	Sampling Year	No of Samples	% of Positive Samples	Range of Concentration (μg/kg)	Reference
Wheat	DON	2011	51	65	115–275	[15]
	ZEN		69		7–107	
	FUMs		39		28–203	
	T-2		25		16–18	
Barley	DON		34	53	74–228	
	ZEN		9		5–68	
	FUMs		15		25–121	
	T-2		32		5–26	
Oat	DON		33	21	34–201	
	ZEN		6		4–43	
	FUMs		6		25–31	
	T-2		18		5–10	
Wheat	T-2/HT-2	2014–2015	61	32	20.5–56.2	[68]
Oat			49	69	11.6–304.2	
Barley			23	30	32.2–82.5	
Wheat	AFB1	2015–2016	25 */27 **	3/4	2.1–2.4	[77]
Oat			13/14	7/8	1.6–1.7	
Barley			11/13	8/-	2.2	
Rye			7/9	11/-	2.1	
Wheat	OTA		25 */27 **	4/16	1.9–3.2	
Oat			13/14	14/15	1.8–2.0	
Barley			11/13	-/23.1	1.8–3.2	
Rye			7/9	33/27	2.1–2.9	
Wheat	DON		25 */27 **	59/56	27–1220	
Oat			13/14	57/61	32–546	
Barley			11/13	61/54	32–389	
Rye			7/9	22/57	31–113	
Wheat	ZEN		25 */27 **	56/60	4.7–115	
Oat			13/14	36/46	3.2–63	
Barley			11/13	46/36	3.3–36.8	
Rye			7/9	22/27	3.6–7.8	
Wheat	FUMs		25 */27 **	26/40	39–215	
Oat			13/14	21/38	35–52	
Barley			11/13	54/27	43–126	
Rye			7/9	33/29	35.6–74	
Wheat	DON AFs, FUMs, ZEN, T-2, HT-2, OTA	2016 -	57 -	39 -	63–867 -	[78]
Wheat	AFB1	2017	47	2	16.2	
	AFB2			2	4.4	
	AFG1			2	20.9	
	AFG2			2	2.2	
	DON			43	68–2408	
	HT-2			23	3–7	
	OTA			8	0.3–614	
	ZEN			-	-	
	T-2			-	-	
Wheat	DON AFs, FUMs, ZEN, T-2, HT-2, OTA	-	57 -	68 -	63–867 -	[79]
Wheat	Ergot alkaloids	2021	64	2	68	[80]
Rye			11	18	76–167	

*—conventionally cultivated cereals; **—organically cultivated cereals.

**Table 3 foods-13-01391-t003:** Occurrence of mycotoxins in meat products in Croatia.

Type of the Product	Mycotoxin	No of Samples	% of Positive Samples	Range of Concentration (μg/kg)	Reference
Fermented meat products	OTA	90	64	1.23–7.92	[105]
	CIT		4	1.0–1.3	
	AFB1		10	1.0–3.0	
Traditional pork meat products	AFB1	410	to 11	to 1.69	[28]
	OTA		to 20	to 9.95	
Dry-fermented sausages	AFB1	24	41	1.62– 4.49	[26]
Slavonian kulen	OTA	99	41	1.82–17.00	[27]
Dry cured and fermented meat products	OTA	187	22	1.36–9.95	[35]
“Slavonski kulen”	AFB1	45	22	1.84–14.46	[28]
	OTA		36	1.97–19.84	
Traditional meat products, prosciutto and sausages	AFB1	160	8	to 1.92	[32]
	OTA		14	to 6.86	
Dry sausages	OTA	88	15	to 0.48	[33]
	AFB1		0	<LOD	
Dry sausages	CPA	47	15	2.55–59.80	[34]
Traditional and industrial dry fermented sausage *Kulen*	CPA	26	19	to 13.35	[31]
	OTA		27	to 6.95	
	AFB1		<LOD	<LOD	
Dry fermented meat products	AFB1	250	<LOD	<LOD	[30]
	OTA		10	to 4.81	
	STC		4	to 3.93	
	CPA		13	to 335.5	
	CIT		<LOD	<LOD	

**Table 4 foods-13-01391-t004:** Occurrence of mycotoxins in other food from Serbia.

Type of the Product	Mycotoxin	Sampling Year	No of Samples	% of Positive Samples	Range of Concentration (μg/kg)	Reference
Breakfast cereals	AFB1	2012	82	-	-	[134]
		2015	54	11	0.06–0.15	
	OTA	2012	82	21	0.07–11.81	
		2015	54	13	0.09–2.33	
Peanuts and peanut-based products	AFB1	2017–2019	94	19	0.25–13.10	[129]
Maize milling products and maize-based products			117	21	0.28–28.15	
Tree nuts			125	38	0.21–3.36	
Rice			41	22	0.17–1.60	
Millet			14	57	1.03–2.27	
Mixed composition products			41	32	0.56–1.55	
Dried figs			25	52	1.01–1.51	
Dried ground red paprika			6	33	0.5–2.88	
Infant fruit juice	Patulin	2013–2015	48	44	1.6 ± 2.2 *	[135]
Infant purée			66	17	0.6 ± 1.5 *	
Juices for children			100	43	2.4 ± 5.2 *	
Apple juice	Patulin	2013–2015	73	74	6.4 ± 10.6 *	[136]
Multi-fruit juice			69	28	2.1 ± 4.9 *	
Bilberry juice	Patulin	2021	8	1	3 **	[137]
Black chokeberry juice			8	-	-	
Wine	OTA	2011	8	38	0.018	[138]
		2012	17	35	0.024	
		2013	30	27	0.020	
		2014	31	71	0.035	
		2015	15	87	0.030	
		2016	12	58	0.018	
Different pepper types (red pepper powder, white and black pepper)	AFS OTA	2010	17	-	-	[139]
Nuts (walnut, hazelnut, almond, peanut)	AFS	2013	17	-	1.20–3.48	[140]
	OTA			-		
	ZEN			12		
	T-2			-		
	HT-2			-		
	FB1			-		
	FB2			-		
Biscuits with fruit filling	AFS	2014	39	13	AFB1 0.91–1.92	[141]
	OTA			44	AFB1 0.91–1.92	
	DON			-	-	
	ZEN			51	AFB1 0.91–1.92	
	T-2			36	AFB1 0.91–1.92	
	HT-2			-	-	
Cookies	AFS		34	-	-	
	OTA			3	18.3	
	DON			-	-	
	ZEN			-	-	
	T-2			-	-	
	HT-2			-	-	
Dried fruit	AFS		14	-	-	
	OTA			-	-	
	DON			-	-	
	ZEN			-	-	
	T-2			-	-	
	HT-2			-		
Fruit jams	AFS		10	30	AFB1 0.93–4.17	
				10	AFB2 1.15	
	OTA			30	12.0–25.0	
	DON			-	-	
	ZEN			-	-	
	T-2			-	-	
	HT-2			-	-	

* Mean ± Standard deviation (μg/kg). ** μg/L.

**Table 5 foods-13-01391-t005:** Occurrence of mycotoxins in other food from Croatia.

Type of the Product	Mycotoxin	Sampling Year	No of Samples	% of Positive Samples	Range of Concentration (μg/kg)	Reference
Bread and other bakery products	DON	-	111	17	-	[142]
White wine Red wine	OTA	2018	19	63	0.269–0.496 *	[143]
			15	67	0.254–0.565 *	
Apple juice	Patulin	2014–2016	122	21	8.57 **	[144]

* μg/L; ** Medium value.

## Data Availability

The original contributions presented in the study are included in the article; further inquiries can be directed to the corresponding author.

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
