# Peer review of "Climate Change and Mycotoxins Trends in Serbia and Croatia: A 15-Year Review"

_foods, 2024, doi:10.3390/foods13091391_

Round 1
Reviewer 1 Report
Comments and Suggestions for Authors
This article describes the impact of climate change on mycotoxin contamination. Climatic conditions are a key factor in the production of mycotoxins in cereals, and a systematic analysis of the relationship between the two will help to establish an early warning system for mycotoxin risk and provide a basis for the development of management measures. However, there are some problems with this article.
1, this article is too much content, can be worth focusing on the analysis of mycotoxin contamination in cereals.
2、This paper just describes the climate data and mycotoxin contamination data, and there is no correlation analysis between the two, this part should be supplemented, otherwise, it is not in line with the theme.
3、The conclusion part is too much and should be shortened.
Comments on the Quality of English Language
Moderate editing of English language required
Author Response
Reviewer 1
The authors would like to express their sincere gratitude to the Reviewer for his constructive and useful comments, as well as his professional review. They believe that this contribution will greatly enhance the quality of this Manuscript and will significantly contribute to its improvement and comprehensive enhancement. The authors have made every effort to incorporate these valuable suggestions into the Manuscript, and they hope that in its current form, it meets the criteria for publication in the journal Foods.
REVIEWER COMMENT: This article describes the impact of climate change on mycotoxin contamination. Climatic conditions are a key factor in the production of mycotoxins in cereals, and a systematic analysis of the relationship between the two will help to establish an early warning system for mycotoxin risk and provide a basis for the development of management measures. However, there are some problems with this article. This article is too much content, can be worth focusing on the analysis of mycotoxin contamination in cereals.
AUTHORS COMMENT:
Thank you for your comment. The authors acknowledge the Reviewer's observation regarding the extensive nature of the article. Nevertheless, the Authors hold a strong belief that such comprehensiveness was essential to adequately encapsulate the scope of research published over the past 15 years from both Serbia and Croatia pertaining to mycotoxins occurrence in food as well as a review of weather conditions during this period. However, there is disagreement regarding the reviewer's suggestion to solely focus the article on results related to cereals. Presenting the results as a comprehensive analysis covering different types of food makes the paper unique, incorporating more than 120 contemporary references, as there is currently no other published article that comprehensively combines available data in this manner. Therefore, to the best of our knowledge, this is the FIRST REVIEW PAPER that combines fifteen years of results from two different countries. The primary goal of this UNIQUE REVIEW PAPER is to provide readers access to results from a 15-year period across two countries and to gain insight into mycotoxin occurrence, weather conditions, and trends in one place.
REVIEWER COMMENT: This paper just describes the climate data and mycotoxin contamination data, and there is no correlation analysis between the two, this part should be supplemented, otherwise, it is not in line with the theme.
AUTHORS COMMENT:
Thank you for your comment. As previously stated, the aim of this review paper was to present available data regarding the occurrence of mycotoxins in food from Croatia and Serbia, as well as to conduct an analysis of weather conditions over a 15-year period. The results related to mycotoxins occurrence were gathered from other published studies, while the analysis of weather data for 15 years was conducted by the authors. Since we obtained the results regarding the occurrence of mycotoxins from other studies, correlating them with weather conditions would be very difficult or even impossible. This is due to variations in the presentation of data on mycotoxin occurrence, which differ or lack detail across different studies. For example, the differences among available results regarding the occurrence of mycotoxins include variations in the number of samples, sampling periods, analytical methods applied (such as ELISA, HPLC, LC-MS/MS), different limits of quantification. Furthermore, various statistical methods were used for results analysis across different papers. Additionally, some results are presented as average values, while others are presented as ranges, percentages, etc. All of the above indicates that it would be almost impossible to establish a correlation between weather conditions and the occurrence of mycotoxins based on the differently presented results from numerous studies. Moreover, the title of the paper does not indicate that a correlation between weather conditions and mycotoxins occurrence was conducted. Therefore, it is not entirely clear how the reviewer perceives that without correlation analysis, the paper is not in line with the theme. If necessary, the title of the Manuscript can be changed.
REVIEWER COMMENT: The conclusion part is too much and should be shortened.
AUTHORS COMMENT: Thank you for your comment. The conclusion has been shortened.
REVIEWER COMMENT: Moderate editing of English language required
AUTHORS COMMENT: Thank you for your comment. The first version of the paper has already been reviewed by a proficient English user. In response to the reviewer's comments, the entire paper has been further reviewed by another proficient English user, and several changes and/or improvements have been made accordingly.
Reviewer 2 Report
Comments and Suggestions for Authors
The conclusions of the manuscript show that meteorological data reveal a warming trend with increasing temperatures and more frequent drought conditions, especially pronounced during the summer months. These changes foresee the occurrence of an impact on the occurrence of mycotoxins in agricultural raw materials and foodstuffs in Serbia and Croatia. However, the manuscript has some internal inconsistencies that should be resolved.
The title and abstract of the manuscript do not clearly reflect the scope and limitations of the study. The title refers to a multi-year review of the effect of climate change on mycotoxin trends in Serbia and Croatia; however, the development is based on meteorological changes occurring over a 15-year period. It is suggested to adapt the title and abstract to the content of the manuscript.
Meteorological variations in Serbia and Croatia are presented in Figures 1 and 3 by representative lines of average precipitation and air temperature in 1981-2010 and visually compared against observed values in the periods 2009-2023. It is suggested to use a probabilistic "band" or "channel" of the historical behavior of these variables and compare the observed values with a statistical method to determine which years were outliers.
It is suggested to make a synthesis effort (as in Table 1) to elaborate other tables with corn, other cereals and milk that concentrate the descriptive values (range, average, proportion) reported for mycotoxins, which are organized according to the periods and places analyzed.
It is suggested to perform a statistical significance analysis to show more precisely the magnitude of the differences. It is also highly recommended to try to perform a meta-analysis with the available data to show more clearly and precisely the association between meteorological changes and the occurrence of toxigenic fungi and their mycotoxins, as well as to try to achieve the goal of providing "a comprehensive overview of trends" of the effect of meteorological fluctuations and mycotoxins occurrence in Serbia and Croatia.
The description of changes in climatic conditions and mycotoxin contamination in Serbia and Croatia is done by very long texts in which comparative adjectives are used to qualify (major/minor, huge/slightly, etc.) the differences found in each season. Therefore, it is suggested to concentrate on a short, clear and unambiguous description of the effects of meteorological changes on mycotoxin occurrence presented in a series of tables.
Author Response
Reviewer 2
The authors would like to express their sincere gratitude to the Reviewer for his constructive and useful comments, as well as his professional review. They believe that this contribution will greatly enhance the quality of this Manuscript and will significantly contribute to its improvement and comprehensive enhancement. The authors have made every effort to incorporate these valuable suggestions into the Manuscript, and they hope that in its current form, it meets the criteria for publication in the journal Foods.
REVIEWER COMMENT: The conclusions of the manuscript show that meteorological data reveal a warming trend with increasing temperatures and more frequent drought conditions, especially pronounced during the summer months. These changes foresee the occurrence of an impact on the occurrence of mycotoxins in agricultural raw materials and foodstuffs in Serbia and Croatia. However, the manuscript has some internal inconsistencies that should be resolved.
The title and abstract of the manuscript do not clearly reflect the scope and limitations of the study. The title refers to a multi-year review of the effect of climate change on mycotoxin trends in Serbia and Croatia; however, the development is based on meteorological changes occurring over a 15-year period. It is suggested to adapt the title and abstract to the content of the manuscript.
AUTHORS COMMENT: Thank you for your comment. Given your concern, the authors provide the following explanation: Studying changes in precipitation and temperature over a 15-year period (2009–2023) and comparing them with the long-term average values (1981–2010) falls under the investigation of climate change. Research covering such a timeframe and analyzing trends relative to long-term averages aims to identify changes in climatic conditions over time. This is characteristic of studying climate change as it focuses on analyzing long-term trends in weather conditions, as opposed to meteorological changes, which typically refer to short-term fluctuations in weather conditions. To avoid any confusion for both the Reviewer and the readers, the authors have decided to improve the title of the paper in “Climate Change and Mycotoxins Trends in Serbia and Croatia: A Multi-Year 15-Year Review”.
REVIEWER COMMENT: Meteorological variations in Serbia and Croatia are presented in Figures 1 and 3 by representative lines of average precipitation and air temperature in 1981-2010 and visually compared against observed values in the periods 2009-2023. It is suggested to use a probabilistic "band" or "channel" of the historical behavior of these variables and compare the observed values with a statistical method to determine which years were outliers.
AUTHORS COMMENT: Thank you for your comment. The Figures 1 and 3 represent the total seasonal precipitation and average seasonal air temperature for each of the 15 examined growing seasons in the period 2009-2023 for Serbia and Croatia, respectively. In Figures 1 and 3, the average values for the period of 1981–2010 are shown with the aim of illustrating the anomalies of air temperature and precipitation in the last 15 seasons in comparison to the long-term period. Based on the figures, it is very easy to observe that both countries exhibit a warming trend, as in almost every year in Serbia, the recorded temperature was above the values from the long-term period, while in Croatia, the temperature value was close to the average in only a few years. Additionally, the authors consider that it is very noticeable from the figures that none of the 15 examined growing seasons had an air temperature below the long-term average value. Regarding precipitation, the authors also believe that based on the figures, it can be easily observed in which years a deficit or excess of precipitation was noted. To further understand the trends in weather conditions, as well as extremes, additional Figures 2 and 4 were created, depicting monthly air temperature anomalies and dry/wet conditions which are ranging from exceptionally dry to normal to exceptionally wet. Based on all of the above, the authors believe that Figures 1-4 clearly represent the weather conditions for each of the 15 years (2009-2023), as well as extreme seasons. In addition, creating new visuals, which include 'band' or 'channel,' would be a time-consuming process that requires additional data to be downloaded, processed, and prepared.
REVIEWER COMMENT: It is suggested to make a synthesis effort (as in Table 1) to elaborate other tables with corn, other cereals and milk that concentrate the descriptive values (range, average, proportion) reported for mycotoxins, which are organized according to the periods and places analyzed.
AUTHORS COMMENT: Thank you for your comment. Based on the reviewer comments, the 4 new tables were prepared and implemented in the paper:
Table 1. Occurrence of mycotoxins in cereals from Serbia
Table 2. Occurrence of mycotoxins in cereals from Croatia
Table 4. Occurrence of mycotoxins in other food from Serbia
Table 5. Occurrence of mycotoxins in other food from Croatia
With the addition of the previous Table (earlier Table 1, now Table 3) "Occurrence of mycotoxins in meat products in Croatia," the revised paper now contains 5 Tables. The authors have made significant effort for Tables preparation. In addition to these 5 tables, the authors have decided not to transfer the remaining data into table format for several reasons. As can be noticed, although there is the least available data regarding the occurrence of mycotoxins in cereals (excluding maize), other food categories, and meat, these tables are extensive and large. Furthermore, despite the results being presented in table form, the text and discussion on the results, as well as weather conditions for the examined years, cannot be completely omitted. Therefore, the inclusion of tables did not lead to the reduction in the overall length of the paper. Based on this, it can be argued that the tables regarding results for milk and maize would be significantly more extensive, with the discussion of results, weather conditions, authors' conclusions, and other aspects also unable to be removed. Therefore, the desired reduction in the length of the paper would not be achieved. Furthermore, it is difficult to create a table and uniformly present the results because, for example, in the case of maize, one study examines the presence of 1 mycotoxin, while another study examines more than 100 mycotoxins. Additionally, the authors used different statistical tools for data processing, resulting in the presentation of results in various ways. Some results reflect situations by seasons, some by years, while there are other differences in how results are presented in different studies. Based on all the aforementioned, the authors have decided to present some results in 5 tables, while other results are presented in text form, allowing for a brief overview of results, discussion, and conclusion for each study.
The authors want to emphasize that this paper aims to provide a comprehensive overview and present results from two countries over a 15-year period, incorporating more than 120 contemporary references. While acknowledging the extensive nature of this review paper, the Authors strongly believe in its necessity to encapsulate research from both Serbia and Croatia on mycotoxin occurrence in food and weather conditions during this period. Presenting the results as a comprehensive analysis covering various food types makes this PAPER FIRST AND UNIQUE, as no other published article combines available data in this manner. Therefore, this review paper stands as the first to integrate fifteen years of results from two countries, with the primary goal of providing readers with comprehensive insights into mycotoxin occurrence, weather conditions, and trends in one place
REVIEWER COMMENT: It is suggested to perform a statistical significance analysis to show more precisely the magnitude of the differences. It is also highly recommended to try to perform a meta-analysis with the available data to show more clearly and precisely the association between meteorological changes and the occurrence of toxigenic fungi and their mycotoxins, as well as to try to achieve the goal of providing "a comprehensive overview of trends" of the effect of meteorological fluctuations and mycotoxins occurrence in Serbia and Croatia.
The description of changes in climatic conditions and mycotoxin contamination in Serbia and Croatia is done by very long texts in which comparative adjectives are used to qualify (major/minor, huge/slightly, etc.) the differences found in each season. Therefore, it is suggested to concentrate on a short, clear and unambiguous description of the effects of meteorological changes on mycotoxin occurrence presented in a series of tables.
AUTHORS COMMENT: Thank you for your comment. The aim of this review paper was to present available data regarding the occurrence of mycotoxins in food from Croatia and Serbia, as well as to conduct an analysis of weather conditions over a 15-year period. The results related to mycotoxins occurrence were gathered from published studies, while the analysis of weather data for 15 years was conducted by the authors. Since we obtained the results regarding the occurrence of mycotoxins from other studies, it is not possible to correlate them with weather conditions, as the data on mycotoxins occurrence vary or are insufficiently detailed in different studies. For example, the differences among available results regarding the occurrence of mycotoxins include variations in the number of samples, sampling periods, analytical methods applied (such as ELISA, HPLC, LC-MS/MS), different limits of quantification. Furthermore, various statistical methods were employed for results analysis across different papers. Additionally, some results are presented as average values, while others are presented as ranges, percentages, etc. All of the above indicates that it would be almost impossible to establish a correlation between weather conditions and the occurrence of mycotoxins based on the differently presented results from numerous studies. Moreover, the title of the paper does not indicate that a correlation between weather conditions and mycotoxin occurrence was conducted. Therefore, it is not entirely clear how the reviewer perceives that without correlation analysis, the paper is not in line with the theme. If necessary, the title of the Manuscript can be changed.
Round 2
Reviewer 1 Report
Comments and Suggestions for Authors
This manuscript has been revised in the light of the comments, it is recommended that it be accepted
Comments on the Quality of English Language
Minor editing of English language required
Author Response
The authors extend their gratitude to the reviewer for their commentary. We are pleased that the reviewer acknowledges the enhancements made to the manuscript following the review process and deems it suitable for publication in the Foods journal.
Reviewer 2 Report
Comments and Suggestions for Authors
The manuscript has been substantially improved and some internal inconsistencies have been resolved. The title and abstract of the manuscript were modified and more clearly reflect the scope and limitations of the study.
A synthesis effort was made to develop additional descriptive tables about mycotoxin reports in the analyzed periods and locations.
The wording of the association between meteorological changes and the occurrence of toxigenic fungi and their mycotoxins was also improved, so that the warming trend, with increasing temperatures and drought conditions, especially pronounced during the summer months, is better appreciated. These changes are associated with the occurrence of mycotoxins in agricultural raw materials and foodstuffs in Serbia and Croatia.
Author Response

(The authors gave the same response as above.)
